# Aerosol hygroscopicity over the South-East Atlantic Ocean during the biomass burning season: Part II – Influence of sea salt and burning conditions on CCN hygroscopicity

Haochi Che[1,2,a], Lu Zhang[2,a], Michal Segal-Rozenhaimer[3,4,a], Caroline Dang[3,4], Paquita Zuidema[5], Arthur J. Sedlacek III[6]

[1]Department of Geosciences, University of Oslo, Oslo, 0315, Norway
[2]Department of Environmental Science, iClimate, Aarhus University, Roskilde, 4000, Denmark
[3]Bay Area Environmental Research Institute, NASA Ames Research Center, Moffett Field, CA 94035, USA
[4]NASA Ames Research Center, Moffett Field, CA 94035, USA
[5]Rosenstiel School of Marine, Atmospheric and Earth Science, University of Miami, Miami, FL 33149, USA
[6]Brookhaven National Laboratory, Upton, NY 11973, USA
[a]Formerly at Department of Geophysics, Tel Aviv University, Tel Aviv, 69978, Israel

Corresponding authors: Haochi Che (haochi.che@geo.uio.no) and Lu Zhang (luzhang@envs.au.dk)

## Abstract

Biomass burning (BB) significantly influences cloud condensation nuclei (CCN) concentrations over the southeastern Atlantic; however, aerosol hygroscopicity ($\kappa$)—a key factor for CCN activation—remains poorly constrained during the BB season. This study investigates $\kappa$ variability using in situ measurements from Ascension Island during the 2016 and 2017 BB seasons. Results show substantial monthly variability, with $\kappa$ values lowest in August and increasing through October. On average, $\kappa$ was significantly higher in 2017 (~0.55) than in 2016 (~0.33), suggesting that the aerosols in 2017 were more hygroscopic and more easily activated as CCN. Sulfate and sea salt were the two dominant contributors to $\kappa$ and the primary drivers of its interannual variability. During the 2017 BB season, sulfate—the major inorganic component—accounted for ~34% of the submicron aerosol mass, while sea salt, estimated via $\kappa$-closure analysis, contributed ~17%. The higher $\kappa$ in 2017 was largely attributed to increased sea salt, likely driven by stronger marine winds. Approximately 67% of sulfate was linked to BB emissions. Variations in BB combustion efficiency, modulated by regional meteorology, influenced sulfate fraction and thus $\kappa$ values. Specifically, higher relative humidity and lower wind speeds over BB source regions in 2017 favored smoldering combustion, explaining the higher sulfate fraction. Overall, the observed interannual differences in aerosol hygroscopicity reflect the combined impacts of BB combustion characteristics and sea salt emissions, underscoring the critical roles of both BB and marine aerosol sources in regulating aerosol-cloud interactions over the southeastern Atlantic.

# 1 Introduction

The Southeastern Atlantic Ocean (SEA) is covered by one of Earth's most extensive stratocumulus cloud decks (Wood, 2012). These semi-permanent clouds can result in a significant radiative effect on global climate (Soden and Vecchi, 2011; Wood, 2012). A distinctive feature of these stratocumulus clouds is their interaction with biomass burning (BB) aerosols transported from widespread fires in southern Africa from June to October. Although these aerosols substantially impact the underlying stratocumulus cloud deck and the overall radiative balance through aerosol-cloud interactions, the specific effects remain poorly understood, contributing to uncertainties in climate models (Adebiyi and Zuidema, 2016; Che et al., 2021; Gordon et al., 2018; Wilcox, 2012).

A critical aspect of aerosol-cloud interactions involves aerosol particles acting as cloud condensation nuclei (CCN), which facilitate the formation of cloud droplets under supersaturated conditions. Variations in CCN concentrations can significantly impact cloud properties and precipitation patterns (Boucher et al., 2013; Christensen et al., 2020; Lu et al., 2018; Ramanathan et al., 2001; Rosenfeld et al., 2008). The effectiveness of aerosol particles as CCN is mainly determined by their size and hygroscopicity, with the latter characterized by the parameter $\kappa$, which is influenced by the chemical composition of aerosols (Petters and Kreidenweis, 2007). Changes in $\kappa$ can strongly influence cloud droplet formation, especially in the marine boundary layer (MBL) over the SEA, where intermediate aerosol concentrations (500-800 cm$^{-3}$) prevail (Kacarab et al., 2020).

BB is a major source of aerosol particles transported into the SEA, contributing large quantities of organic aerosol (OA) and black carbon (BC). The chemical composition of BB aerosol particles is strongly dependent on combustion conditions and the extent of aging they undergo, leading to uncertainties in $\kappa$ values (Akagi et al., 2012; Hodshire et al., 2019). Laboratory studies have found that freshly emitted BB aerosol particles exhibit a broad range of $\kappa$ values (from 0.06 to 0.6), but after a few hours of aging, $\kappa$ converges to approximately $0.2 \pm 0.1$ (Engelhart et al., 2012). This change in $\kappa$ with aging is primarily driven by the oxidation of OA and the production of secondary organic aerosols, while the initial variation in $\kappa$ in fresh BB aerosols is related to the proportion of inorganic components (Engelhart et al., 2012). Additionally, photolysis has been found to significantly reduce OA mass during the weeklong aging of BB aerosols over the SEA, thus affecting their hygroscopicity (Dobracki et al., 2025; Sedlacek et al., 2022). Furthermore, BB aerosols within the SEA boundary layer may undergo cloud processing during their transport through extensive stratocumulus clouds from the free troposphere, further altering their chemical and physical properties (Che et al., 2022a). Therefore, aerosol hygroscopicity $\kappa$ can have large variations due to the multiple processes affecting the chemical composition of BB aerosols in the SEA, potentially resulting in substantial uncertainties in aerosol-cloud interactions.

In addition to BB aerosols, sea-salt particles also significantly affect CCN concentrations in the marine boundary layer over the SEA because of their high $\kappa$ values (Dedrick et al., 2024; Petters and Kreidenweis, 2007). A study examining collected particles in the SEA found that BB and sea-salt aerosol particles are generally well mixed in the marine boundary layer (Dang et al., 2022). Consequently, the hygroscopicity of aerosols in this region varies with changes in the relative contributions of

BB aerosols and sea salts. While a review suggests that a $\kappa$ value of ~0.7 ± 0.2 is useful as a first approximation for remote ocean environments (Andreae and Rosenfeld, 2008), this value is likely too high for the SEA, where a BB aerosol presence reduces overall $\kappa$ values in the mixed aerosol population.

The single scatter albedo of aerosols observed within the MBL in the SEA showed significant changes throughout the BB season (Zuidema et al., 2018b), which are likely driven by variations in BB combustion conditions (Che et al., 2022b; Dobracki et al., 2025). Different combustion conditions can lead to differences in the chemical and physical properties of BB aerosols (Akagi et al., 2012; Zheng et al., 2018), resulting in corresponding differences in the overall $\kappa$ of aerosols in the MBL during the BB season. However, there are few studies focusing on the hygroscopicity of aerosols in the MBL in the SEA, and most of these studies have examined only short time durations (Kacarab et al., 2020; Maßling et al., 2003). Consequently, large uncertainties remain in the overall aerosol hygroscopicity and its variability, affecting cloud properties and the regional climate.

The Layered Atlantic Smoke Interactions with Clouds (LASIC), a 17-month field observation campaign, took place from June 1, 2016, to October 31, 2017, on Ascension Island in the SEA to address these uncertainties. The island lies midway between Africa and South America, within the trade wind shallow cumulus regime, and is frequently impacted by BB aerosol plumes from southern Africa during the BB season (Adebiyi and Zuidema, 2016). The continuous observations from LASIC enable the investigation of changes in aerosol hygroscopicity during the BB season. Filter samples were collected during a flight aircraft campaign (The CLoud–Aerosol–Radiation Interaction and Forcing, CLARIFY) near Ascension Island in 2017 to provide more detailed information on aerosol mixing state and chemical components (Dang et al., 2022). In the current study, we used CCN observations from the island to derive the hygroscopicity $\kappa$ and analyze its monthly variation during the BB season. Additionally, we used filter samples to investigate the contributions of marine and BB emissions to the overall $\kappa$ observed on Ascension Island.

## 2 Method

### 2.1 In-situ field observations

The LASIC campaign was carried out at the Atmospheric Radiation Measurement (ARM) First Mobile Facility (AMF1) site on Ascension Island, located at latitude -7.97°, longitude -14.35°, and an altitude of 341 m. A detailed description of the sampling location and instruments is provided in the campaign report (Zuidema et al., 2018a). Here, we offer a brief overview of the data and instruments used in this study. The CCN concentrations at fixed supersaturations were measured using a cloud condensation nuclei counter (CCNC, Droplet Measurement Technologies, model CCN-200) (Roberts and Nenes, 2005). Aerosol size distributions, ranging from diameters of 10 nm to 500 nm, were measured with a Scanning Mobility Particle Spectrometer (SMPS, TSI, model 3936). Concentrations of aerosol chemical components—including organics, sulfate, nitrate, ammonium, and chloride—were measured using a quadrupole aerosol chemical speciation monitor (Q-ACSM, Aerodyne

Research). Refractive black carbon (BC) was measured using a Single Particle Soot Photometer (SP2, Droplet Measurement Technologies), and carbon monoxide (CO) concentrations were measured using the CO/N2O/H2O Analyzer (CO-ANALYZER, Los Gatos Research, model 098-0014). All instruments were calibrated and the data converted to standard temperature and pressure conditions. Additionally, BC data after September 1, 2017, were omitted from the analysis due to an inlet issue that affected the accuracy of SP2 measurements. Aerosol chemical composition data are available only for 2017 because of instrumentation issues in 2016. The ratio of black carbon to excess carbon monoxide (BC/$\Delta$CO) was calculated to assess BB combustion conditions, where $\Delta$ represents the enhancement above background levels. Background CO concentrations were determined monthly, defined as the 5th percentile of measured CO values for each month. The BC/$\Delta$CO ratio is unitless, as both BC and CO were converted to the same units of mass concentration. To minimize the influence of instrument noise under clean conditions, we only included data where BC mass concentrations were greater than 20 ng m$^{-3}$ in the BC/$\Delta$CO calculations. As a result, clean atmospheric conditions were not considered in analyses involving BC/$\Delta$CO. A more detailed discussion about BC/$\Delta$CO can be found in Che et al. (2022b).

## 2.2 CCN and SMPS correction

The aerosol number size distributions measured by SMPS expressed as dN/dlogD, were fitted with a bimodal log-normal distribution. This fitted distribution was then extrapolated to obtain the integrated aerosol number concentration (N$_{SMPS}$) from 10 to 1000 nm. Since the SMPS measurement range is limited to 10–500 nm, the size distribution between 500 and 1000 nm was obtained through extrapolation. To assess the validity of this extrapolation, we validated the results with measurements from the Ultra-High Sensitivity Aerosol Spectrometer (UHSAS), which covers a broader size range from 60 to 1000 nm. As shown in Figure S1 (Supplement), UHSAS data confirmed that aerosol concentrations above 500 nm were generally low and lacked distinct peaks, supporting the extrapolation.

The CCN concentration at 1% supersaturation was generally up to ~ 30% higher than N$_{SMPS}$, indicating potential counting discrepancies between the instruments. These discrepancies may lead to an overestimation of the CCN activation rate and, consequently, an overestimation of the derived $\kappa$. Therefore, it is necessary to correct it before calculating $\kappa$. A common correction method assumes that at high supersaturation (e.g., 1%), all particles become activated into cloud droplets; thus, the N$_{SMPS}$ and the CCN number concentration at 1 % supersaturation (N$_{CCN1\%}$) should match. By fitting the N$_{CCN1\%}$ to N$_{SMPS}$ and scaling the CCN or SMPS dataset with the fitted parameters, the counting discrepancy can be corrected. However, this assumption may overestimate CCN concentration, particularly when a large fraction of aerosol particles falls within the nucleation mode. For instance, according to the $\kappa$-Köhler equation, the activation of a 20 nm particle at 1% supersaturation requires a $\kappa$ value of approximately 2.24, which is unrealistically high given that most aerosols have hygroscopicity values less than 1 (Petters and Kreidenweis, 2007). During the LASIC campaign, the aerosol number distribution frequently exhibited a bimodal mode, with the smaller mode having a mode diameter of approximately 40 nm (Dobracki et al., 2025). This indicates

a high fraction of small aerosol particles (below 40 nm) observed on Ascension Island. As these smaller particles are less likely

to activate at 1% supersaturation, assuming full activation of all aerosols can significantly overestimate $\kappa$.

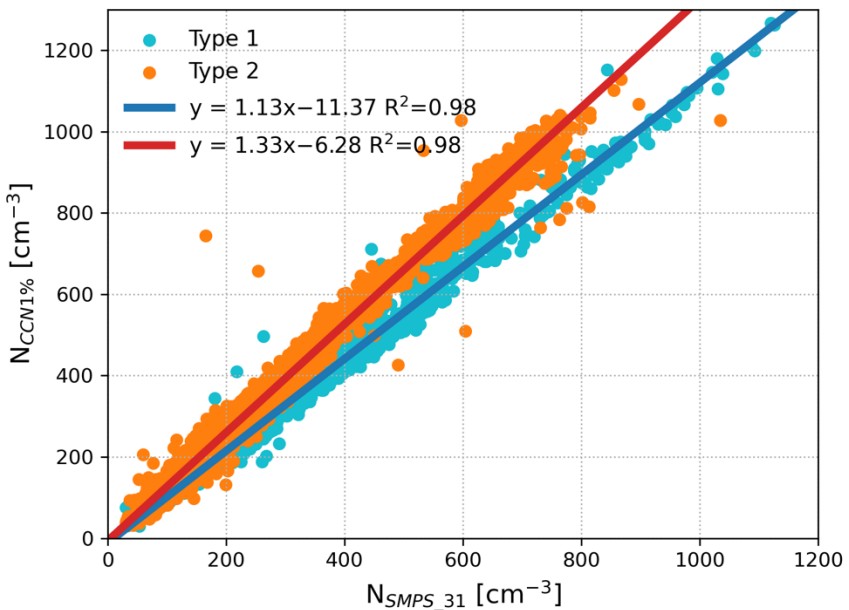

Figure 1. Correlations between the CCN concentration measured at 1% supersaturation ($N_{CCN1\%}$) and the integrated aerosol concentration with diameters from 31 to 1000 nm based on SMPS measurements ($N_{SMPS\_31}$). Two distinct correlations were identified in the data using the K-means clustering method, indicated by light blue and orange points. Type 1 (light blue)

includes data from June, July, September, and October of 2016, while Type 2 (orange) includes data from August 2016 and June to October 2017. The blue and red lines represent linear regressions fitted to each correlation type, respectively.

Here, we propose a modified method to correct CCN and SMPS counting discrepancies by assuming that only particles above a certain critical activation diameter are activated at 1% supersaturation. The key step is to determine the critical activation

diameter of CCN at this supersaturation. Based on the assumption that aerosols at Ascension Island consist primarily of highly aged BB particles and sea-salt aerosols, we assigned a representative $\kappa$ of 0.6 for particles in the lower part of the Aitken mode range (~ 20-40 nm)—similar to that of sulfate aerosols. According to $\kappa$-Köhler theory, this corresponds to a critical activation diameter of 31 nm at 1% supersaturation. We then compared $N_{CCN1\%}$ with the integrated aerosol concentration $N_{SMPS\_31}$ from 31 to 1000 nm. This approach reduces the likelihood of overestimating $\kappa$ by excluding nucleation-mode particles unlikely to

activate, but it may still result in a slight overestimation, as particles smaller than 31 nm typically contain a higher organic fraction and may have $\kappa$ values lower than 0.6. However, sensitivity tests demonstrate that adjusting the assumed $\kappa$ value for

calculating the initial diameter in the $N_{SMPS}$ integration has minimal impact on our results, as detailed in Supplementary Notes 1.

Applying our modified approach, we observed two types of relationships between $N_{CCN1\%}$ and $N_{SMPS\_31}$ during the BB seasons in 2016 and 2017. We used K-means clustering to quantitatively categorize and separate the data, with the results shown in Figure 1. Type 1 includes observations from the 2016 BB season, excluding August, while Type 2 includes observations from the 2017 BB season and August 2016. Both types exhibit strong linear relationships with $R^2 \sim 0.98$. Although the underlying cause of these distinct relationships remains uncertain, the shift in correlation patterns may be related to instrumental changes

in the CCN counter, as further discussed in Supplementary Note 2. After the two clusters were identified, we derived separate linear fits for each type and used the corresponding equations to scale the CCN data, thereby correcting for counting discrepancies between CCN and SMPS.

## 2.3 Calculation of $\kappa$

The aerosol hygroscopicity $\kappa$ in our study was calculated using SMPS and the corrected CCN data through the following

procedure. First, the aerosol size distribution from the SMPS was fitted to a bimodal log-normal distribution. Starting with a particle diameter D = 31 nm, we extrapolated this fitted size distribution and integrated it from D to 1000 nm. When the integrated aerosol number concentration matched the observed CCN concentration, the current particle diameter D was considered the critical diameter $D_c$. If there was no match, we continued increasing D by 1 nm and repeated the steps until a match was found. Once $D_c$ was identified, the hygroscopicity parameter $\kappa$ was calculated using the following equation (Eq. 1),

which is an analytical approximation valid primarily for $\kappa > 0.2$ (Petters and Kreidenweis, 2007):

$$\kappa = \frac{4A^3}{27D_c^3 ln^2(S_c + 1)} \qquad (1)$$

where $A = \frac{4M_w\sigma_{s/a}}{RT\rho_w}$, $M_w$ is the molecular weight of water, $\sigma_{s/a}$ is the surface tension of the droplet/air interface and equals 0.072 J m$^{-2}$, $R$ is the universal gas constant, $\rho_w$ is the density of water, $T$ is the temperature and equals 298 K, and $S_c$ is the supersaturation in the unit of %. Because cloud supersaturation in this region is generally low and mostly below 0.1% (Che et

al., 2021), we used the CCN concentration measured at 0.1% supersaturation to estimate $\kappa$ in this study.

## 2.4 Sea salt estimation

To estimate the contribution of sea salt (represented as NaCl) to the observed aerosol hygroscopicity, we conducted a $\kappa$-closure analysis by comparing the $\kappa$ values derived from CCN measurements ($\kappa_{CCN}$) with those calculated from aerosol chemical composition ($\kappa_{chem}$). The $\kappa_{chem}$ values were computed using aerosol chemical composition data from the Q-ACSM (for non-

refractory components) and SP2 (for BC), applying the volume-weighted mixing rule described by Petters and Kreidenweis (2007). The detailed densities and $\kappa$ values of individual aerosol components used in the calculation are listed in Table S1 (Supplementary Note 3).

The closure procedure involved the following steps:

a. We first derived the volume fractions of each species based on measured aerosol chemical composition, following the simplified ion-pairing scheme proposed by Gysel et al. (2007), and applying Eq. (2):

$$\varepsilon_i = \frac{M_i/\rho_i}{\sum_j M_j/\rho_j} \qquad (2)$$

where $\varepsilon_i$ is the volume fraction, $M_i$ is the mass concentration, and $\rho_i$ is the density of species $i$.

b. The initial $\kappa_{chem}$ was calculated with volume-weighted contributions of all species and their corresponding $\kappa$ values,
applying the Zdanovskii-Stokes-Robinson (ZSR) mixing rule, as in Eq. (3):

$$\kappa_{chem} = \sum_i \varepsilon_i \kappa_i \qquad (3)$$

where $\kappa_i$ is the hygroscopicity parameter for species $i$. Since NaCl was not measured directly, its initial volume was set to zero.

c. We then computed the difference between the $\kappa_{CCN}$ and $\kappa_{chem}$, referred to as the $\kappa_{residual}$ ($\kappa_{residual} = \kappa_{CCN} - \kappa_{chem}$), to estimate
how much sea salt was needed.

d. If $\kappa_{residual}$ exceeded 0.02 (equivalent to 1% of the lowest campaign-wide $\kappa_{CCN}$), small increments of NaCl volume (1% of the total non-NaCl aerosol volume) were iteratively added. After each addition, $\kappa_{chem}$ was recalculated, and the $\kappa_{residual}$ was re-evaluated.

e. This iterative adjustment continued until the $\kappa_{residual}$ was reduced to 0.02 or less. The final amount of added NaCl was then
considered the estimated sea salt contribution required to achieve closure with the observed $\kappa_{CCN}$.

## 2.5 Filter samples

Filter samples were collected using Facility for Airborne Atmospheric Measurements (FAAM) filter systems (Sanchez-Marroquin et al., 2019) on the UK's Bae-146 aircraft near Ascension Island in August 2017. Samples were deposited on Paella TEM grids and analyzed with a JEOL™ JEM-2010F FEG-TEM equipped with a ThermoNoran™ energy-dispersive X-ray
detector (EDX) at Tel-Aviv University. There were 14 samples collected near the island, but only 6 taken inside the MBL were used for further analysis in this study. A detailed description of each sample is provided by Dang et al. (2022). A total of 231 particles were analyzed, and elemental mass fractions for individual particles were determined through EDX analysis. Based on back trajectory analyses, filters collected in the MBL were evenly mixed between BB and marine sources, ensuring a

representative mix of aerosol sources in the study. A list of detected elements, along with their mean mass fractions and standard deviations, is presented in Table S2 (Supplement).

## 2.6 Reanalysis data

Monthly mean winds at a height of 10 meters above the surface, and air relative humidity at 2 meters above the surface over Africa and the SEA during the BB season for the years 2016 and 2017 were analyzed in this study using data from the ERA5 reanalysis dataset, the fifth-generation reanalysis of the global climate and weather, produced by the European Centre for Medium-Range Weather Forecasts (ECMWF) (Hersbach et al., 2020). The dataset has a horizontal resolution of $0.25° \times 0.25°$.

Monthly mean aerosol data (2-dimensional) from the Modern-Era Retrospective Analysis for Research and Applications version 2 (MERRA-2, Global Modeling and Assimilation Office GMAO, 2015) were used to evaluate changes in sea salt mass fraction. The analysis focused on the surface mass concentration of aerosol components within the PM2.5 (particles smaller than 2.5 μm). The MERRA-2 dataset provides a horizontal spatial resolution of $0.5° \times 0.625°$.

 **3 Results**

## 3.1 Variation of aerosol hygroscopicity and chemical composition

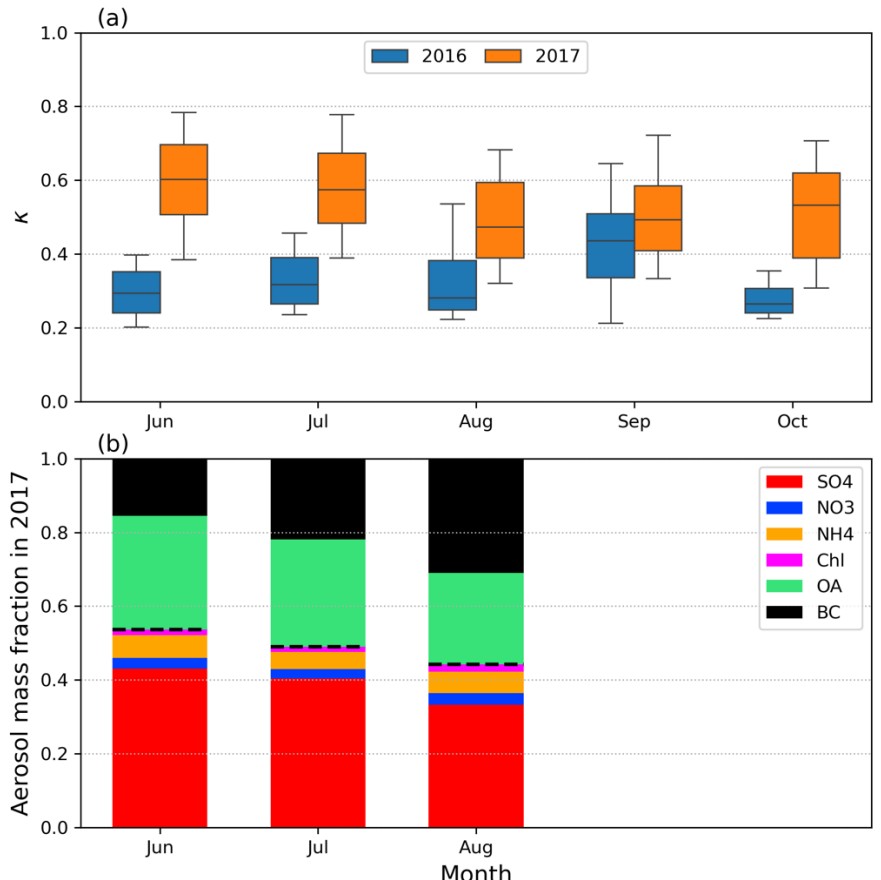

Figure 2. (a) Monthly distributions of $\kappa$ values calculated at 0.1% supersaturation during the BB seasons of 2016 and 2017. The boxes represent the 25th, 50th (median), and 75th percentiles, while the whiskers extend to the 10th and 90th percentiles. (b) Monthly mean aerosol chemical mass fractions measured by ACSM and SP2 during the 2017 BB season. The black dashed line in (b) represents the separation between inorganic and organic fractions of the aerosol components.

The hygroscopicity parameter $\kappa$ exhibited considerable monthly differences throughout the BB season, with pronounced differences between the years 2016 and 2017 (Figure 2a). The monthly changes in $\kappa$ closely aligned with the patterns observed in aerosol absorption coefficients, characterized by a decline from July to a minimum in August, followed by an increase in September and October (Zuidema et al., 2018). This trend underscores the significant influence of biomass combustion on the hygroscopic properties of marine boundary layer aerosols in the SEA. The monthly values of $\kappa$ in 2016 ranged from 0.21 to 0.65, with a mean of 0.33. In contrast, values in 2017 ranged from 0.3 to 0.78, with a mean of 0.55. According to Andreae and

Rosenfeld (2008), $\kappa = 0.7 \pm 0.2$ serves as a typical value for marine aerosols, and Schulze et al. (2020) reported $\kappa$ values around 0.7 for remote marine regions in the Pacific. The significantly lower $\kappa$ values observed in this study suggest a strong influence of African BB on the SEA boundary layer, deviating from the characteristics of a typical remote marine environment during the BB season. This conclusion is supported by the ORACLES (ObseRvations of Aerosols above CLouds and their intEractionS) 2017 flight campaign over the SEA (Redemann et al., 2021), which reported average $\kappa$ values for marine boundary layer aerosols ranging from 0.2 to 0.4 (Kacarab et al., 2020).

The distinct difference in $\kappa$ values between 2016 and 2017 likely reflects changes in aerosol chemical composition between these two years. This is supported by Figure 2(b), which shows the monthly mean mass fractions of aerosol compositions during the BB season in 2017 measured by Q-ACSM and SP2. The mean $\kappa$ values generally aligned with the dashed line representing the fraction of inorganic components. In general, inorganic aerosol components exhibit higher $\kappa$ values due to their greater ability to uptake water vapor (Petters and Kreidenweis, 2007). Consequently, the lower $\kappa$ values observed during the 2016 BB season might be attributed to a reduction in inorganic aerosol components and/or an increase in organic aerosols compared to 2017. These findings suggest substantial changes in the marine boundary layer aerosol chemical properties in the SEA between 2016 and 2017, despite the same general sources of aerosols—primarily BB and sea salt. Additionally, variations in the hygroscopicity of OA between the two years might have contributed to the differences in $\kappa$. However, since the $\kappa$ values for OA are generally low (Pöhlker et al., 2023; Zhang et al., 2023), such changes may not fully account for the overall lower $\kappa$ values observed in 2016.

For the inorganic components, a plausible explanation for the observed differences in $\kappa$ between 2016 and 2017 could be changes in sulfate mass fraction. As illustrated in Figure 2(b), sulfate aerosols constituted the primary inorganic component, accounting for approximately 39% of the total aerosol mass and around 80% of the inorganic aerosol mass on average during the BB season in 2017. Furthermore, as illustrated in Figure S10, the mean sulfate mass fraction increased with $\kappa$, ranging from 0.2 to 0.6, as $\kappa$ rose from 0.3 to 0.75. Given that sulfate aerosols have a relatively high $\kappa$ value of ~0.6 (Petters and Kreidenweis, 2007), changes in sulfate mass fraction might significantly affect the overall $\kappa$ observed on the island.

In addition to sulfate, sea salt (NaCl) is another key contributor that may help explain the high $\kappa$ values observed in 2017. Sea salt has a high hygroscopicity, with its $\kappa$ values reaching up to 1.5 (Zieger et al., 2017). Due to its refractory nature, NaCl cannot be directly measured by the Q-ACSM, making its contribution difficult to quantify. However, the observation of aerosol $\kappa$ values exceeding 0.6—the typical value associated with pure sulfate aerosols—throughout all months of the 2017 fire season provided strong indirect evidence for the presence of sea salt. Notably, the mean sulfate mass fraction under these high-$\kappa$ conditions remained between 0.5 and 0.6 (Figure S10), indicating that sulfate alone cannot account for the observed hygroscopicity. Given that NaCl is significantly more hygroscopic than sulfate, even modest increases in its mass fraction can lead to substantial changes in overall $\kappa$. Therefore, a slightly higher proportion of sea salt is a plausible explanation for the increased aerosol hygroscopicity observed in 2017.

## 3.2 Sources of sulfate aerosols

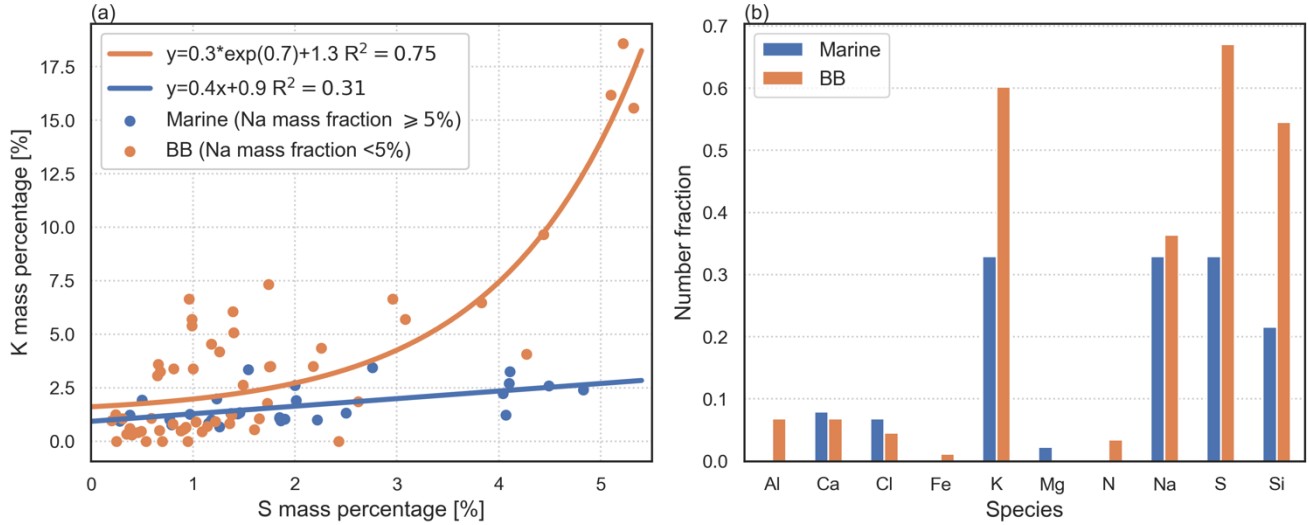

Figure 3. (a) Mass percentage of sulfate and potassium from different potential sources of aerosols, and (b) number fraction of elements in sulfate-bearing particles. Particles originating from BB and marine sources are differentiated by the sodium mass fraction; blue represents marine origin, and orange represents BB origin. The lines in (a) represent the fitted correlations, with the corresponding equation and $R^2$ value displayed in the legend.

Sulfate aerosols observed at Ascension Island likely originate from three different sources: transported BB emissions, other anthropogenic activities, and local marine sources (Hossain et al., 2024). The marine contribution to sulfate arises mainly from the oxidation of dimethyl sulfide (DMS) emitted from the ocean surface, while BB and anthropogenic sources contribute through the oxidation of sulfur dioxide released during combustion. Notably, much of the BB activity in Africa is itself anthropogenic, as evidenced by a pronounced Sunday minimum pattern—a clear signature of human influence tied to weekly activity cycles (Earl et al., 2015). During the fire season, BB emissions are expected to dominate over other anthropogenic sulfate sources. However, since both BB and non-BB anthropogenic sulfate are transported from continental regions and often arrive together, distinguishing their individual contributions remains challenging and requires additional research. For this study, we categorized all sulfate from BB and non-BB anthropogenic sources during the fire season as primarily of BB origin. This classification is supported by Figure 3 (a), which revealed a strong correlation between sulfate and potassium for BB-origin aerosols.

Here, we used the data from the single-particle filter analysis conducted by Dang et al. (2022) to examine the primary sources of sulfate. A potential way to distinguish the contributions of marine and BB sources to sulfate is by using specific tracers for

these sources. Potassium (K) is commonly considered a tracer of BB emissions, while sodium (Na) is predominantly found in marine emissions (Seinfeld and Pandis, 2016). Although Na was also present in BB plumes over the SEA, it was in lower concentrations and likely accounted for a smaller weight percentage in individual BB aerosols when analyzed by Energy Dispersive X-ray Spectroscopy (EDX) (Dang et al., 2022). Therefore, we selected K and Na as tracers for BB aerosols and marine emissions and made a simple distinction between the sulfate sources based on their concentrations.

In Figure 3, a Na mass fraction threshold of 5% has been set to distinguish between the two sulfate sources. This value has been determined iteratively to allow for the separation of the two sources. When the Na mass percentage exceeded 5%, we considered marine to be the predominant aerosol source; conversely, when it was below 5%, BB aerosols were considered dominant. As shown in Figure 3(a), K and sulfate exhibited a strong exponential relationship in aerosols when BB was the dominant source, possibly because the excess K was bonded with nitrates (Dang et al., 2022). This suggests that an increase in BB aerosols can significantly elevate the proportion of sulfate. For aerosols where marine was the main source (blue markers in Figure 3a), there was no evident relationship between sulfate and K, resulting in a near-zero slope in the fitted line. Both correlations had p-values lower than 0.05, indicating statistical significance. Therefore, these distinct correlations confirm that the method for distinguishing sulfate origins was reliable and successful in providing a general indication of the dominant source.

Using this classification method, we categorized and quantified all sulfate-bearing particles sampled within the boundary layer near Ascension Island, as presented in Figure 3(b). Both K and Na were present in aerosol particles from both sources, with number fractions exceeding 30%. This is because the observed BB aerosols were well mixed after week-long aging and transportation (Che et al., 2022a), consistent with the results from single-particle analysis by Dang et al. (2022) in this region. Nevertheless, K remained predominantly concentrated in BB-origin aerosols. While the number of Na-containing particles was similar between the two sources, their mass fractions differed—marine aerosols exhibited a higher Na mass fraction (Figure 3a). The results indicate that BB was the main source of observed sulfate, with approximately 67% of sulfate-bearing particles originating from BB. The marine source likely contributed the remaining 33% of sulfate, roughly half of the BB contribution. This conclusion is supported by model simulations from Hossain et al. (2024), which estimated that marine sulfate accounted for around 40% of sulfate near the surface during the fire season. Consequently, BB was the largest source of sulfate observed on the island. Given that DMS emissions are relatively stable from June to October (Hossain et al., 2024), while BB-emitted $SO_2$ showed much greater variability in the SEA region, the observed changes in sulfate mass fraction were most likely due to fluctuations in BB emissions.

**3.3 Effect of BB conditions on aerosol hygroscopicity**

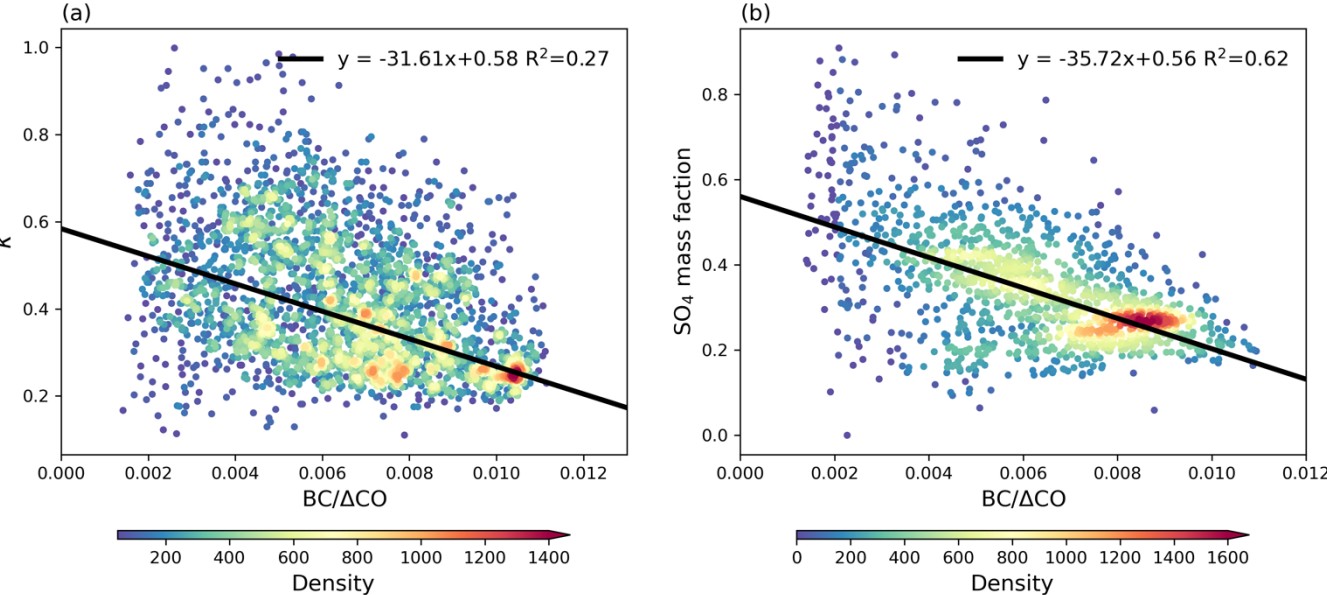

Figure 4. Relationships of BC/ΔCO with (a) $\kappa$ calculated at 0.1% supersaturation and (b) sulfate mass fraction. The black lines represent linear regressions, with the corresponding equations displayed in the legend. The color scale indicates the data density, which is the count of the data in the gridded 50x50 bins of the data range. Note that panel (a) includes data from both the 2016 and 2017 BB seasons, whereas panel (b) only includes data from the 2017 BB season. Data identified as clean conditions, defined by BC mass concentrations below 20 ng m$^{-3}$, have been excluded.

From the previous section, we established that BB was the dominant source of sulfate. Here, we further analyzed how BB influenced the sulfate mass fraction, with the first step being to quantify the burning conditions. A recent study on LASIC aerosols (Che et al., 2022b) found that seasonal variations in aerosol optical properties are affected by BB combustion conditions. These conditions are approximated by BC/ΔCO, with higher BC/ΔCO indicating more flaming combustion, and Δ denotes the difference between observed concentrations and background levels. The value of BC/ΔCO showed a strong linear correlation with modified combustion efficiency (MCE) for freshly emitted BB aerosols, establishing 0.004 as the threshold for distinguishing between flaming and smoldering-dominated combustion in freshly emitted BB aerosols (Che et al., 2022b). For the highly aged aerosols observed on Ascension Island, the threshold of BC/ΔCO was estimated to be approximately 0.003. When BC/ΔCO < 0.003, the BB aerosols observed were considered to be emitted mainly from smoldering combustion; conversely, when BC/ΔCO > 0.003, they were mainly from flaming combustion.

The relationship between BC/ΔCO and $\kappa$ is shown in Figure 4(a). $\kappa$ generally showed a decreasing trend with increasing BC/ΔCO, confirming that BB combustion conditions affected the aerosol hygroscopicity observed at Ascension Island. As BC/ΔCO increases, indicating that BB becomes more flaming-dominated, there is a higher emission of BC (Conny and Slater, 2002). Freshly emitted BC particles are insoluble and can significantly reduce aerosol hygroscopicity. However, BC particles observed at Ascension Island, which went through weeklong aging during their transport, became coated with soluble materials (Che et al., 2022b). Such coating can mitigate the reduction in $\kappa$ due to BC particles, aligning with field observations that show no significant difference in $\kappa$ between coated BC particles and BC-free particles (Ohata et al., 2016). Therefore, the changes in $\kappa$ due to BC/ΔCO are likely driven by changes in the fraction of inorganic and organic components resulting from different burning conditions.

This can be confirmed by Figure 4(b), which shows that the sulfate mass fraction is well correlated with BC/ΔCO, indicating that combustion conditions can influence the sulfate mass fraction in BB aerosols. The correlation is particularly evident when BC/ΔCO > 0.003. This suggests that in flaming-dominated combustion, the sulfate mass fraction decreases as the combustion becomes more flaming. Figure S11 illustrates that both the mass concentration of sulfate and the total aerosol particles increase with BC/ΔCO, implying that the decreasing sulfate mass fraction with higher BC/ΔCO is not due to a reduction in sulfate mass. Instead, it results from a more significant increase in the mass of species such as BC and OA, which decreases the relative fraction of sulfate. When BC/ΔCO < 0.003, indicating smoldering-dominated combustion, the correlation between the sulfate mass fraction and BC/ΔCO weakens due to more scattered data. The aerosol mass was also relatively low when BC/ΔCO < 0.003, suggesting relatively clean conditions. As a result, the contribution of sulfate from other sources, such as marine, became more prominent, leading to a weaker correlation between sulfate mass fraction and BB combustion conditions. This is supported by a study that showed different aerosol size distributions when the BB plume mixes with clean air (Dobracki et al., 2025).

## 3.4 Changes in BB conditions

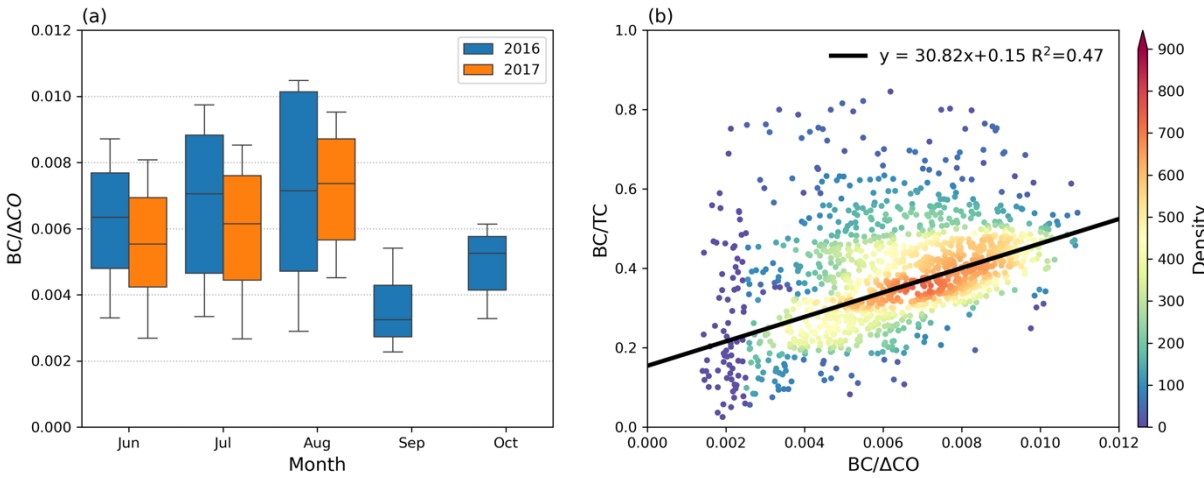

Figure 5. (a) Monthly distributions of BC/ΔCO for the BB seasons of 2016 (blue) and 2017 (orange). The boxes represent the
345    25th, 50th (median), and 75th percentiles, while the whiskers extend to the 10th and 90th percentiles. (b) Relationship between
BC/ΔCO and the BC to TC (total carbon, the sum of OA and BC) mass ratio during the BB season in 2017. The color scale in
(b) indicates data density, represented as the count of data points within gridded 50x50 bins of the data range. Only data with
concurrent CCN measurements are included to minimize sampling bias and ensure consistency across the dataset.

350    The monthly distribution of BC/ΔCO during the BB seasons of 2016 and 2017 is illustrated in Figure 5(a). Given that BC/ΔCO
was negatively correlated with both the sulfate aerosol mass fraction and $\kappa$, the monthly variations in BC/ΔCO suggest changes
in BB conditions that may account for the observed changes in aerosol hygroscopicity. From June to August 2017, an increase
in BC/ΔCO coincided with a decrease in $\kappa$ and sulfate mass fraction. In 2016, considerable fluctuations in BC/ΔCO, as
demonstrated by the large range between the 10th and 90th percentiles in each month, hindered the identification of a clear
355    pattern between the monthly mean $\kappa$ and BC/ΔCO. However, the highest average $\kappa$ in 2016 was found in September, coinciding
with the lowest average BC/ΔCO for that month, suggesting that combustion conditions can explain the observed changes in
$\kappa$ for that year.

The median BC/ΔCO ranged from 0.0032 to 0.0071 in 2016 and from 0.0055 to 0.0073 in 2017. Comparing BC/ΔCO values
across the two years, BB fires in 2017 generally exhibited lower BC/ΔCO values, indicating less flaming combustion. This
360    finding is consistent with the higher $\kappa$ values observed in 2017 compared to 2016. Since BC/ΔCO influences $\kappa$ by affecting
the sulfate aerosol mass fraction, we infer that the variations in burning conditions between 2016 and 2017 resulted in

differences in sulfate aerosol mass fractions. Specifically, the higher BC/ΔCO values in 2016 suggest a generally lower fraction of sulfate aerosols in each month during the BB season, leading to lower aerosol hygroscopicity compared to 2017.

Since the observed BB aerosols were highly aged, the value of BC/ΔCO could also be affected by removal processes such as wet and dry deposition during the transport of the BB plume. Wet removal primarily occurs through interactions with cloud droplets and precipitation, which can reduce the BC/ΔCO ratio. Che et al. (2022b) analyzed the cloud fraction and precipitation during the BB season in 2016 and 2017 and found no significant difference between these two years. This suggests that the major difference in $\kappa$ between 2016 and 2017 was not due to changes in wet deposition. For dry deposition, it is mainly influenced by the transport pathway and distance. However, similar transport pathways and distances were found for both years (Che et al., 2022b), indicating that dry deposition was also not responsible for the changes in BC/ΔCO between these two years. Therefore, the changes in BC/ΔCO may be primarily attributed to changes in burning conditions. This conclusion is supported by Figure 5(b), which shows that the ratio of BC to total carbon (TC, the sum of OA and BC) increases with BC/ΔCO. This finding is consistent with previous studies, such as Conny and Slater (2002), which reported that flaming combustion tends to have a high BC/TC ratio, whereas the BC/TC ratio for smoldering combustion is generally very low. Therefore, we conclude that one of the main drivers of changes in aerosol hygroscopicity is the combustion conditions of BB, which affect the sulfate mass fraction in the aerosols observed on the island.

**3.5 Contribution from sea salt**

Apart from sulfate, sea salt is another potential driver of the observed differences in $\kappa$ between the two years. Due to the lack of direct measurements of NaCl, we estimated its contribution by performing a closure analysis, comparing $\kappa$ values derived from CCN measurements with those calculated from aerosol chemical composition, as described in the Methods section.

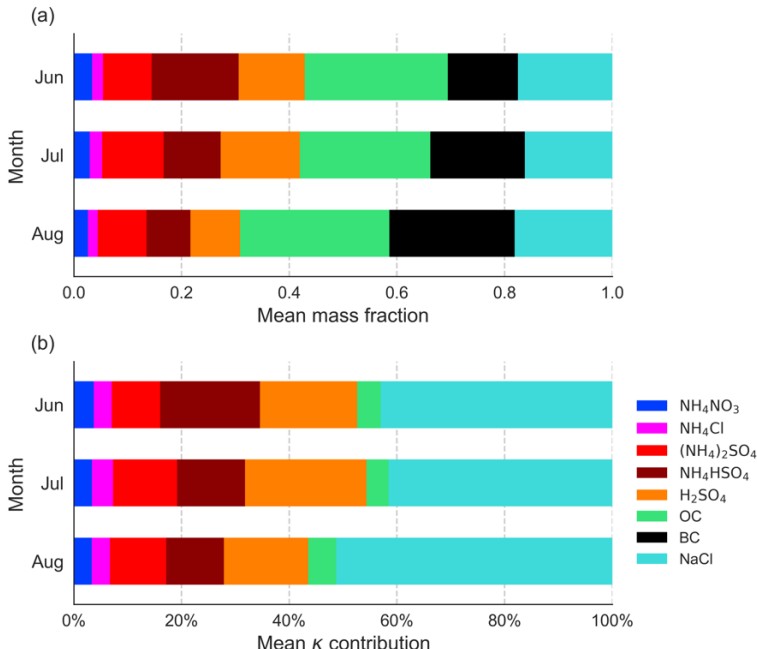

Figure 6. Derived monthly mean (a) mass fractions of aerosol species in submicron aerosols (PM$_1$) and (b) percentage contributions of each aerosol species to the total $\kappa$ during the 2017 fire season.

The results demonstrate that sea salt was a substantial and persistent component of aerosols on Ascension Island (Figure 6a). From June to August 2017, the monthly mean NaCl mass fraction in submicron aerosols (PM$_1$) was approximately 0.17. Even during August 2017, when BB dominated aerosol loading—as indicated by the peak aerosol concentrations and increased fractions of BC and OA—sea salt remained a significant contributor, with a mass fraction of about 0.18. For comparison, during the non-BB season (January to May 2017), sea salt mass fractions ranged from 0.11 to 0.31. The consistently high NaCl fraction suggests that the high $\kappa$ values observed during the fire season cannot be explained by sulfate alone.

Due to measured ammonium (NH$_4^+$) concentrations being insufficient to fully neutralize chloride (Cl$^-$), nitrate (NO$_3^-$), and sulfate (SO$_4^{2-}$), we inferred the presence of sulfuric acid (H$_2$SO$_4$) to account for the excess sulfate ions. However, alternative sulfate salts, such as sodium sulfate (Na$_2$SO$_4$), may also exist. As detailed in Supplementary Notes 3, sensitivity tests exploring different sulfate species showed minimal impact on the estimated sea salt contribution, which consistently remained close to 0.17 in PM$_1$. After accounting for NaCl in the total aerosol mass, the adjusted sulfate mass fraction (representing the sum of all SO$_4^{2-}$-containing species) ranged from 0.37 in June to 0.25 in August, with an average of 0.34.

Monthly mean contributions of each aerosol species to the observed $\kappa$ are shown in Figure 6 (b). NaCl made a substantial contribution, ranging from 43% in June to 51% in August. This was comparable to the contribution from sulfate, which ranged

from 46% to 37%. Together, NaCl and sulfate accounted for about 90% of the total $\kappa$, underscoring their dominant role in shaping aerosol hygroscopicity in the SEA region. Notably, despite their similar influence on $\kappa$, the mass fraction of NaCl was about half that of sulfate, indicating a stronger sensitivity of $\kappa$ to changes in NaCl mass fraction. As a result, even relatively small differences in NaCl mass fraction between 2016 and 2017 could have significantly contributed to the observed differences in $\kappa$. To test the robustness of this conclusion with respect to assumptions about the chemical form of excess sulfate, we recalculated species contributions assuming all excess sulfate was present as $Na_2SO_4$ instead of $H_2SO_4$. Under this alternative scenario, the changes were minimal: NaCl contributions remained essentially unchanged from June to August, while sulfate contributions shifted only slightly, becoming 46%, 48%, and 38%, respectively (Figure S9, and Supplementary Notes 3). This confirms the substantial impact of NaCl on overall aerosol hygroscopicity, even during the BB season in the SEA.

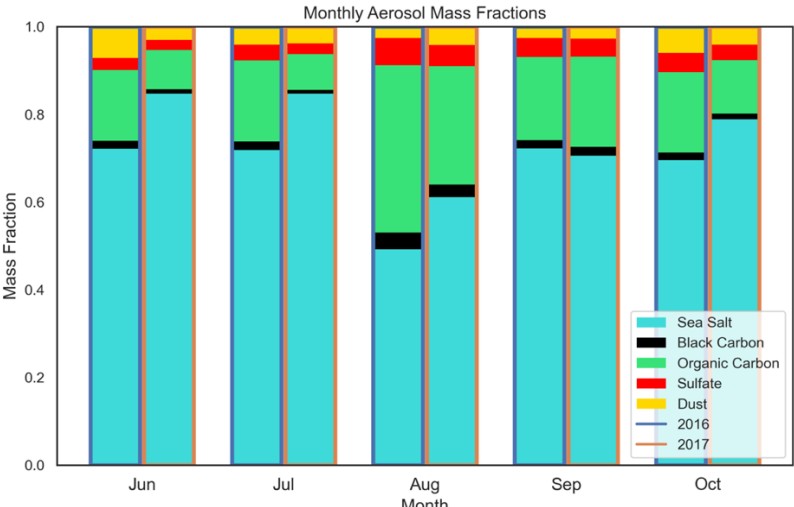

Figure 7. Monthly mean mass fraction of surface aerosol species derived from MERRA-2 reanalysis during the fire seasons of 2016 and 2017. The blue frame highlights the mass fractions in 2016, while the orange frame highlights those in 2017. Note the mass of aerosols below 2.5 um ($PM_{2.5}$) was used.

To investigate the changes in NaCl between 2016 and 2017, we compared monthly sea salt aerosols mass fractions within $PM_{2.5}$ from MERRA-2 reanalysis data during the fire season (Figure 7). Across all months, sea salt mass fractions exceeded 50%, substantially higher than our estimates derived from in-situ measurements restricted to the $PM_1$ size range. A noticeable increase in sea salt mass fraction was observed in 2017, with the most significant increases occurring in June and July. In contrast, the sea salt mass fractions for September remained relatively unchanged between the two years. This trend aligns closely with the changes in $\kappa$, where the largest differences between the two years occurred in June and July, and the smallest

difference was observed in September. Therefore, it is likely that changes in the NaCl mass fraction between the two years contributed significantly to the observed differences in $\kappa$.

### 3.6 Sea salt vs. sulfate in affecting $\kappa$

Since both sea salt and sulfate each contributed approximately 40% to the observed $\kappa$, it is likely that one—or both—of these components was responsible for the significant differences in $\kappa$ between 2016 and 2017. In this section, we examine which of the two was the dominant contributor to the year-to-year variations in aerosol hygroscopicity.

A strong linear correlation between BC/$\Delta$CO and the sulfate mass fraction in 2017 is shown in Figure 4(b). Here, the sulfate mass fraction was calculated as the ratio of observed sulfate to the total measured aerosol mass (no NaCl), thereby minimizing

the influence of sea salt and ensuring that the sulfate fraction primarily reflects BB contributions. Using this correlation, we first estimated the sulfate mass fraction in sea-salt-free (no NaCl) aerosols for 2016 based on the observed BC/$\Delta$CO values from that year. To extend this estimate to the total aerosol population (with NaCl), we applied the linear relationship between sulfate mass fractions with and without NaCl in 2017 (Figure S12). This allowed us to translate the estimated sulfate fractions from sea-salt-free aerosols into corresponding values for NaCl-containing aerosols, which could then be used to calculate

aerosol hygroscopicity.

By comparing these estimated sulfate mass fractions between 2016 and 2017, we quantified the year-to-year differences. Multiplying these differences by the hygroscopicity of sulfate ($\kappa = 0.6$) enabled us to estimate the potential contribution of sulfate variations to the observed $\kappa$ differences between the two years. This procedure probably underestimates the 2016 sulfate fraction, because Figure 7 suggests sea-salt were higher in 2017, which would dilute sulfate in that year relative to 2016.

Nevertheless, the sea-salt increase is less pronounced in PM$_1$ than in PM$_{2.5}$, so the bias in our PM$_1$-based estimate is likely within the uncertainty range evaluated below.

To establish plausible upper and lower bounds for the sulfate mass fraction estimates, we vertically shifted the original linear relationship between sulfate mass fraction and BC/$\Delta$CO (Figure 4b) to encompass the majority of data points (density > 200), as shown in Figure S13. The resulting sulfate mass fractions for NaCl-containing aerosols in 2016 are shown in Figure 8(a).

Averaged over June to August, the sulfate mass fractions ranged from about 0.10 at the lower bound to 0.21 at the median estimate, and up to 0.33 at the upper bound. Although the estimates span a relatively wide range (~0.2) between the highest and lowest scenarios, all consistently exhibit a decreasing trend from June through August.

Using these sulfate estimates for 2016, we calculated the potential contribution of sulfate changes to the observed differences in $\kappa$ between 2016 and 2017, as shown in Figure 8(b). When assuming the highest sulfate mass fraction in 2016, the year-to-

year difference in sulfate is minimal, resulting in a small $\kappa$ contribution. Conversely, the lowest sulfate estimate yields the largest interannual difference, producing the highest $\kappa$ contribution. This range defines the uncertainty bounds for how much of the observed $\kappa$ difference can be attributed to changes in sulfate.

The observed $\kappa$ difference between 2016 and 2017 ranged from approximately 0.3 in June to 0.2 in August. Under the lower-bound estimate, the contribution from sulfate changes to $\kappa$ was minimal—averaging only ~0.02 in June and July (about 10% of the observed difference) and negligible in August. The median estimate yielded an average $\kappa$ contribution of ~0.07 across the three months, explaining roughly 28% of the observed difference. At the upper bound, the sulfate-driven $\kappa$ contribution reached ~0.14, accounting for up to 56% of the total $\kappa$ difference. These results indicate that even under the most favorable assumptions, changes in sulfate mass fraction can explain at most about half of the observed $\kappa$ variation, with the median scenario suggesting a substantially smaller contribution. Therefore, sulfate is unlikely to be the primary driver of the interannual $\kappa$ differences. Instead, the results point to changes in sea salt as the dominant factor influencing aerosol hygroscopicity between 2016 and 2017 on Ascension Island.

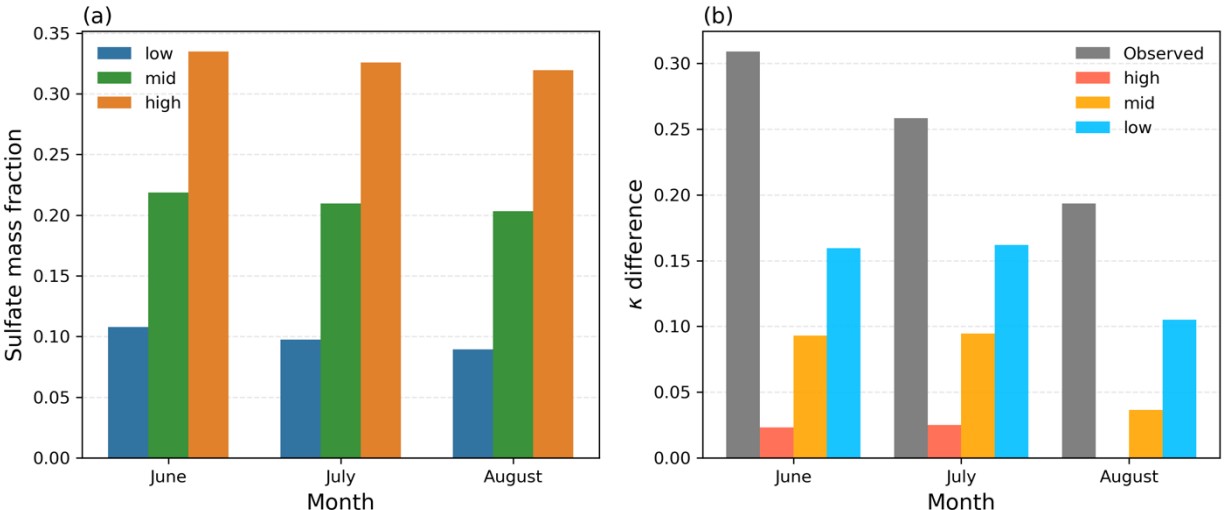

Figure 8. Estimated monthly mean (a) sulfate mass fraction in sea-salt-containing aerosols in 2016 and (b) the $\kappa$ differences between 2017 and 2016 attributable to sulfate variations. In panel (a), the blue, green, and orange bars indicate the range of sulfate mass fraction estimates, representing the lowest, median, and highest possible scenarios, respectively. In panel (b), the gray bars show the observed monthly mean differences in $\kappa$ between the two years, while the colored bars (light blue, orange, and red) depict the $\kappa$ differences explainable by sulfate mass fraction changes under the estimated lowest, median and highest sulfate mass.

## 3.7 Potential changes in burning conditions and sea-salt emissions

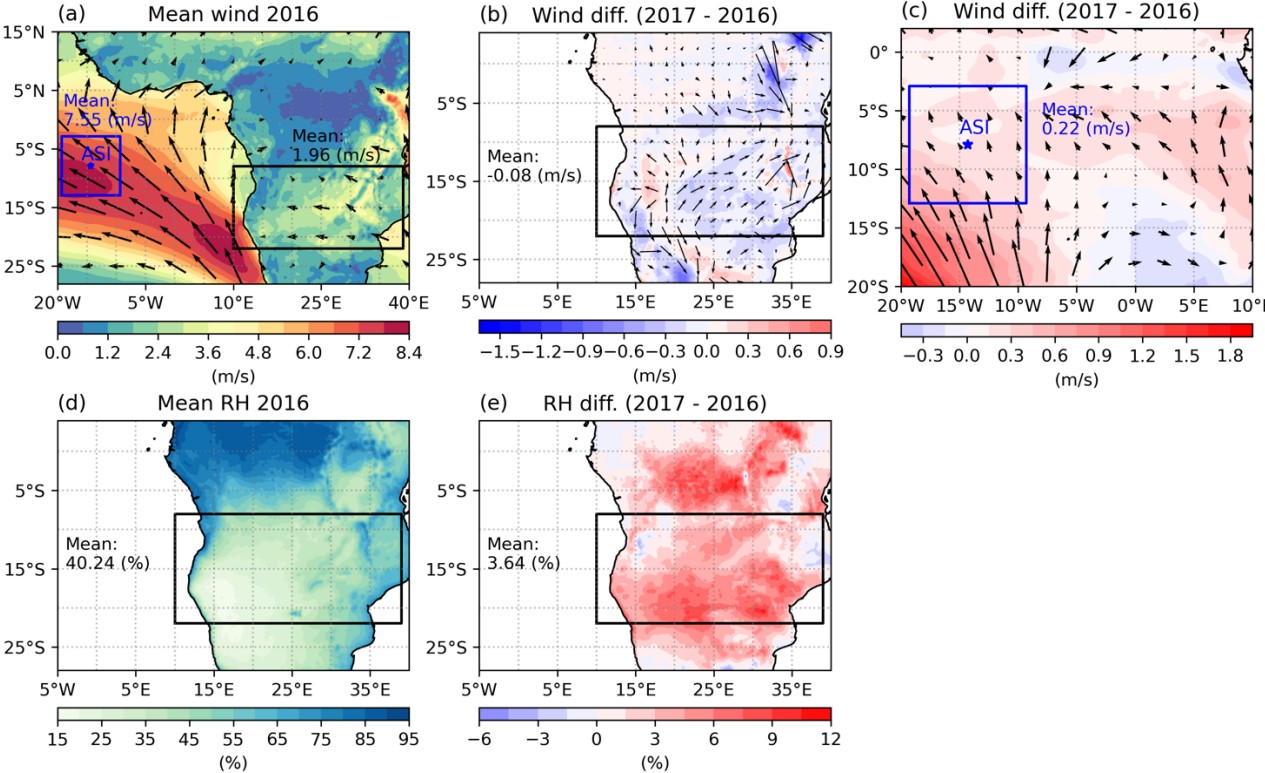


Figure 9. Mean and difference in surface wind and relative humidity (RH) during the BB season (June–October). (a) Mean surface wind in 2016. (b, c) Difference in surface wind between 2017 and 2016 (2017 - 2016) over (b) the African continent and (c) the region surrounding Ascension Island. (d) Mean RH in 2016. (e) Difference in RH between 2017 and 2016 (2017 - 2016). The black boxes indicate the main BB source regions and the blue boxes represent the area centered on Ascension

Island (±4° latitude and longitude). Box-averaged values are shown in the corresponding box colors. Note that in panels (d) and (e), the unit (%) refers to RH values, not relative percentage change.

In the preceding sections, we concluded that the observed interannual changes in $\kappa$ were primarily driven by variations in BB and sea salt, with sea salt playing the dominant role. The influence of BB is mainly attributed to changes in combustion

conditions, while sea salt variations are directly linked to fluctuations in the near-surface wind field (Ovadnevaite et al., 2012). In this section, we examine the meteorological drivers behind these changes, focusing on how atmospheric conditions may influence the combustion state of BB and contribute to enhanced sea salt aerosol production.

BB combustion conditions can be affected by several factors, including fuel moisture content, relative humidity (RH), and wind speed. Here we focus primarily on wind speed and RH. Wind speed directly influences the oxygen supply, which is closely related to combustion efficiency, while RH primarily affects burning conditions by influencing the moisture level.

The mean wind field is shown in Figure 9(a). Easterly winds dominated over both the land and Ascension Island. During the 2016 BB season, the mean wind speed was approximately 7.6 m/s around Ascension Island and around 2 m/s over the BB source regions on land (excluding the ocean). In 2017, wind speed over the BB regions on land decreased slightly by about 0.08 m/s (Figure 9b), accompanied by an increase in westerly winds, which oppose the direction of plume transport. This shift in wind direction may have reduced the efficiency of BB aerosol transport toward Ascension Island, potentially decreasing the influence of BB aerosols at the site in 2017. In addition, wind speed can influence combustion efficiency: higher wind speeds promote flaming combustion by enhancing oxygen supply, although strong winds can also suppress flaming by increasing cooling (Santoso et al., 2019). Therefore, the slight (~ 4%) reduction in wind speeds during the 2017 BB season may have limited oxygen supply, promoting less flaming combustion. This interpretation is consistent with observed changes in BC/ΔCO. Together, the reduced wind speed and altered wind direction likely contributed to both modified combustion conditions and decreased BB aerosol transport, helping to explain the interannual differences in aerosol composition and hygroscopicity observed at Ascension Island.

The mean RH over the land during the 2016 BB season is shown in Figure 9(d). The average RH across the BB region was approximately 40.2% that year, indicating relatively dry conditions favorable for flaming combustion. In 2017, the mean RH increased to 43.9%, representing a relative increase of about 9% (Figure 9e). Although RH in 2017 still reflected relatively dry conditions, the increase may have reduced the prevalence of flaming combustion and favored smoldering fires by increasing the fuel moisture content. This change in RH is likely another factor contributing to the differences in BC/ΔCO between 2016 and 2017.

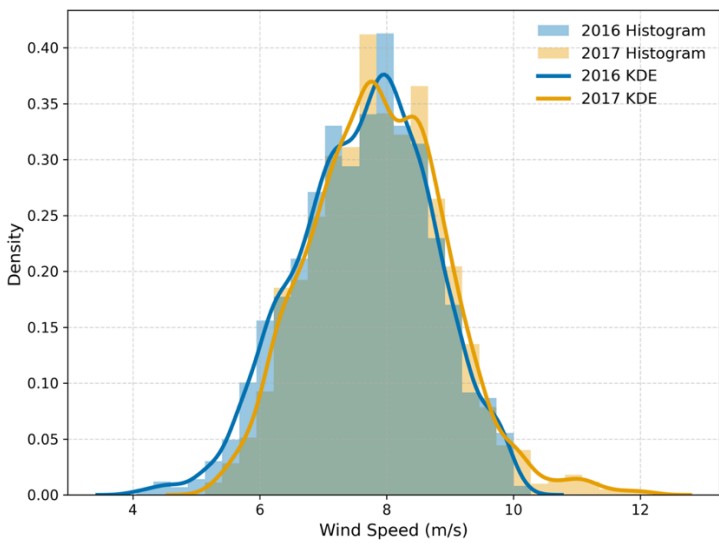


Figure 10. Density distribution of area-weighted mean hourly wind speeds during the BB seasons of 2016 and 2017, calculated for the region surrounding Ascension Island (blue box in Figure 9).

Sea-salt aerosol concentrations are strongly influenced by surface wind speed, following a power-law relationship
(Ovadnevaite et al., 2012). In the region highlighted in blue in Figure 9(c), centered on Ascension Island and spanning ± 4 degrees in both latitude and longitude, mean surface wind speeds during the 2017 BB season increased by approximately 0.22 m/s, or about 3%, compared to 2016. Additionally, the surface wind direction became more southeasterly, indicating enhanced transport of marine air masses from the Southern Atlantic towards the island. These changes likely contributed to an increase in sea-salt aerosol mass in 2017. Although the overall increase in mean wind speed was relatively small, the production of sea
salt aerosols is highly sensitive to wind speed, particularly under higher wind regimes. Figure 10 shows that the frequency of wind speeds exceeding 10 m/s more than doubled in 2017, supporting a notable enhancement in sea salt emissions. Given the high hygroscopicity of sea salt, even modest increases in its mass fraction can significantly impact in overall aerosol hygroscopicity.

Overall, meteorological conditions during the BB season showed notable regional differences between 2016 and 2017. Over
the major BB regions in southern Africa, surface wind speeds slightly decreased, and RH increased in 2017. These conditions favor smoldering over flaming combustion, consistent with the observed reduction in BC/ΔCO ratios and the inferred increase in sulfate mass fractions. At the same time, sea surface wind speeds around Ascension Island increased, promoting enhanced marine air mass transport and greater sea-salt aerosol production. These findings collectively highlight the role of regional

meteorological changes—both in BB source regions and in the marine environment surrounding Ascension Island—in shaping

aerosol composition and hygroscopicity.

## 4 Conclusion and discussion

From June 1, 2016, to October 31, 2017, an *in-situ* observation field campaign was conducted on Ascension Island. In this study, we calculated the aerosol hygroscopicity parameter $\kappa$ using CCN and aerosol size distribution data to investigate aerosol hygroscopicity during the BB season (June-October) in 2016 and 2017.

The $\kappa$ values exhibited significant monthly changes throughout the BB season, decreasing from June and July to the lowest values observed in August and an increase from September to October. Around 90% of the $\kappa$ values ranged from 0.21 to 0.78, with most values falling below the commonly used $\kappa$ value ($0.7 \pm 0.2$) for marine aerosols, indicating a strong influence of BB on the marine boundary layer in the SEA. A noticeable difference was observed between the BB seasons of 2016 and 2017, with a mean $\kappa$ of approximately 0.33 in 2016 and about 0.55 in 2017. This significant difference suggests that aerosols in 2017

were more hygroscopic and more readily activated into cloud droplets under the same supersaturation.

The alignment between monthly mean $\kappa$ values and the fraction of inorganic components suggests that variations in $\kappa$ were largely driven by changes in aerosol chemical composition. In 2016, lower $\kappa$ values can be attributed to a reduction in inorganic content and a higher organic fraction. Sulfate was the dominant measured inorganic species, contributing ~39% of the total aerosol mass and ~80% of the inorganic mass in 2017, with monthly values ranging from 33% to 43%. This indicates sulfate

played a key role in driving $\kappa$ variability. However, our measurements excluded refractory species such as sea salt (NaCl), which can significantly contribute to overall $\kappa$ due to its high hygroscopicity. To estimate sea salt's contribution, we conducted a $\kappa$ closure analysis by comparing $\kappa$ values derived from CCN measurements with those calculated from aerosol chemical composition, while iteratively adjusting the assumed sea salt mass fraction. The results suggest that sea salt accounted for ~17% of the aerosol mass, reducing the adjusted sulfate mass fraction (including NaCl) to ~34%. Although mass fraction of sea salt

was only about half that of sulfate in submicron aerosols, its contribution to $\kappa$ was roughly equal (~45%) due to its higher intrinsic hygroscopicity. Together, sulfate and sea salt accounted for approximately 90% of total aerosol hygroscopicity, making them the two most influential species controlling $\kappa$ on the island. Therefore, the substantial interannual differences in aerosol hygroscopicity are likely driven by changes in the fractions of these two components.

To understand the source and variability of sulfate aerosols, we performed a simple source attribution analysis using filter

samples collected near the island. The results indicate that approximately 67% of sulfate-bearing particles originated from BB emissions, while the remaining 33% came from marine sources. This suggests that changes in BB emissions are likely the primary drivers behind the monthly differences in sulfate mass fraction. To further investigate how BB emissions influence sulfate mass fraction, we quantified the combustion conditions using the BC/$\Delta$CO ratio, where a higher ratio indicates more

flaming combustion. BC/$\Delta$CO showed a negative linear correlation with both $\kappa$ and sulfate mass fraction, indicating that as biomass combustion becomes more flaming, both the sulfate mass fraction and $\kappa$ value tend to decrease. This decrease is attributed to a more significant increase in the mass of species like BC and OA, which reduces the relative fraction of sulfate.

A comparison of BC/$\Delta$CO ratios during the BB season in 2016 and 2017 revealed consistently lower values in 2017, suggesting less flaming combustion that year. This finding aligns with the higher $\kappa$ values observed in 2017, implying that changes in combustion conditions contributed to the interannual differences in aerosol hygroscopicity. However, further analysis showed that even the maximum plausible change in sulfate mass fraction between the two years could explain only about 50% of the observed difference in monthly mean $\kappa$. This suggests that increased sea salt played a dominant role in enhancing aerosol hygroscopicity in 2017 compared to 2016. This conclusion is supported by MERRA-2 aerosol reanalysis data, which show that sea salt comprised the majority of aerosol mass in PM$_{2.5}$ and exhibited a notable increase during the 2017 BB season.

The differences in BB combustion conditions and sea salt emissions between 2016 and 2017 appear to be closely linked to regional meteorological changes. Over the southern African BB source regions, slight decreases in surface wind speed and increases in RH during 2017 likely contributed to a shift from more flaming to more smoldering combustion. This shift is consistent with the lower BC/$\Delta$CO ratios and higher sulfate mass fractions observed in 2017. At the same time, surface wind speeds over the ocean near Ascension Island increased slightly, enhancing the transport of marine air masses and contributing to greater sea salt emissions. Given the high $\kappa$ value of sea salt, even a relatively small increase in it mass fraction had a substantial effect on overall aerosol hygroscopicity, leading to a higher mean $\kappa$ in 2017.

In summary, sea salt and BB are the two most influential factors affecting aerosol hygroscopicity over the southeastern Atlantic during the BB season. The large interannual difference in $\kappa$ was primarily driven by changes in sea salt, with additional contributions from variations in BB combustion conditions. Meteorological changes—both over the BB source regions and in the oceanic environment near Ascension Island—play a key role by influencing fire behavior and sea salt emissions, thereby modifying aerosol composition and hygroscopicity. Given the strong influence of aerosol hygroscopicity on cloud droplet activation and radiative properties, these findings underscore the importance of both sources in shaping aerosol-cloud interactions in this region.

**Data availability**

All LASIC ARM data are publicly available at https://adc.arm.gov/discovery/#/results/site_code::asi/start_date::2016-06-01/end_date::2017-10-31. The ERA5 data are from the ECMWF Climate data store website: https://cds.climate.copernicus.eu/datasets. The MERRA-2 data can be downloaded from NASA Goddard Earth Sciences (GES) Data and Information Services Center (DISC): https://disc.gsfc.nasa.gov/datasets?project=MERRA-2.

## Author contributions

HC, LZ and MSR developed the concepts and ideas of the paper. MSR, CD, PZ, and AJS III carried out the observations and provided the data. All authors contributed to the analysis of the results. HC wrote the paper with input and comments from all other authors.

## Competing interests

The contact author has declared that none of the authors has any competing interests.

## Acknowledgment

The authors would like to thank the LASIC team for their support. H.C. and M.S. gratefully acknowledge funding for this research from DOE-ASR grant DE-SC0020084. P.Z. gratefully acknowledges support from DOE-ASR grant DE-SC0021250. We thank Dr. Ernie Lewis for his valuable discussions and feedback on the manuscript. We also sincerely thank the two anonymous reviewers for their constructive comments and insightful suggestions.

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
