# Peer review of "Aerosol hygroscopicity over the South-East Atlantic Ocean during the biomass burning season: Part II – Influence of sea salt and burning conditions on CCN hygroscopicity"

_EGUsphere, 2024_

## Referee Comment (RC2)

**Review Report Che et al.: Aerosol hygroscopicity over the Sout-Easy Atlantic Ocean during the biomass burning season: Part II – influence of burning conditions on CCN hygroscopicity**

The authors present the monthly averages of the particle hygroscopicity from Cloud Condensation Nuclei activity measurements at Ascension Island Station in the Marine Boundary Layer of the Southeastern Atlantic for two consecutive Biomass Burning (BB) seasons (2016 & 2017). They explain the observed differences in the monthly mean values with changes in the aerosol particle composition, namely the change in contribution of sulphate (SO4) species. They claim that these composition changes are mainly stemming from changes in the BB conditions on the African continent.

The topic is suitable for this journal and of interest for the atmospheric science community. While I agree with the general direction of their conclusion (BB aerosol has a very important influence on the hygroscopicity in the SEA region), I am not convinced by their specific arguments and interpretation as I specify in my comments below. Some of the authors' claims appear quite bold to me and lack enough supporting evidence from the presented data. I recommend major revisions to present a more comprehensive interpretation of the data.

Major comments

1) Focussing on SO4 contribution to explain the trends – while ignoring NaCl from see salt for most of the discussion.
   This is my biggest concern about this manuscript. The authors user the change in SO4 contribution as the main argument to explain the observed trends within a year and between the two years. They do mention that NaCl from marine sources probably influences the hygroscopicity and even prove the presence of NaCl in Section 3.2. In the paragraph lines 299 – 304, they state that the main differences between flaming and smouldering burning conditions is indeed the amount of BB aerosol reaching the station. Thus, the burning condition determines the ratio between (more hygroscopic) marine and (very likely less hygroscopic) BB aerosol. For me, this is a key finding but the authors then completely omit the presence of NaCl from their arguments and continue to use the SO4 ratio as the main argument.
   NaCl will affect the observed hygroscopicity much more than SO4 related compounds as it has by far the largest k value of all included species and thus even a small amount will strongly change k. Since the ACSM cannot quantify NaCl, it is impossible to judge how much it is contributing at different times.
   In short, I do not believe that the small change in SO4 contribution can explain the observed differences while ignoring the other constituents – especially NaCl which the ACSM cannot see. In Specific comment (1), I provide more details about my concerns using the approximation of k from the ACSM &SP2 composition to show my doubts about the SO4 fraction argument. The authors need to revise their chain of arguments and broaden their discussion to include the sea salt/NaCl impact and not just focus on SO4.

2) Separating data from clean periods and BB plumes before averaging.
   The authors present most of the data as monthly averages which are, of course, of general interest and should be reported to better understand the average hygroscopicity of this region. (Dobracki et al., 2024) provide an in-depth analysis of composition and optical properties of the 2017 part of the data set. In their Fig 1, they show that there are distinct plumes in the burning season of 2017. From the data in their Table 2, I calculated that the clean air periods account for 5% (September 2017) to 15% (July 2017) of the data. I assume the 2016 data will also show significant differences between the months and potentially also between the years.
   This aspect (the frequency of BB plumes) is not covered in the discussion. Especially for the Section 3.3 and 3.4 it would be beneficial to separate the data from clean periods from the plume ones. In Figs 4 and 5 the clean period data points are probably close to BC/ΔCO=0 and are not distinguished from the smouldering data points with BC/ΔCO<0.003. I would hazard a guess that a lot of the spread of the data points with BC/ΔCO ~0 comes from having both smouldering BB plume data and clean air.

Separating the clean and plume data may also improve the correlation ($R^2$ values are pretty low) and thus strengthen the conclusions drawn from these Figures.

3) I find the approach for the correction of the "counting error" problematic and need some more information and clarifications.
- The authors only show data for the BB seasons (June – October in both years), but their description sounds like that there were continuous measurements for 18 months at the location. How does the rest of the data look like? Was there a continuous drift from Type 1 to Type 2? I.e., was the issue getting gradually worse with time or was there a sudden change?
  A gradual change would point towards drift in instruments and no connection to the aerosol characteristics. A sudden change could point to some event affecting the instruments. If the difference seems chaotic (no clear trends) it is more likely that the discrepancy is linked to some aerosol property that changes over the months (e.g., special features of the size distribution, aerosol type).
- According to Zuidema et al. (2018), there were two more CPCs (cutoff at 10nm and 2.5nm) and a UHSAS at the station. How does the total number concentration from these instruments compare to SMPS and CCNC? Do they also become more different from 2016 to 2017?
- How often were the instruments calibrated during the 18 months? (and which calibrations were performed?)
- I assume there were regular SS calibrations with ammonium sulphate (AS) using the SMPS setup with the same CPC as during the measurements? Was the same concentration mismatch observed in these calibration measurements? Did the mismatch change between 2016 and 2017? Was there a size dependency?
  If the calibrations also showed the discrepancy, it is very likely that there is simply a discrepancy between the CPC and CCNC. TSI states that their CPC counting is +/- 20% directly after calibration. If the calibration data looked fine, there is potentially some other systematic reason for the "counting error" and that it is different in 2016 and 2017.
  If there was a size dependence in the AS data, there was probably a more complex issue than a simple counting error which probably cannot be fixed by a simple correction factor.
- The authors modify the approach by using an activation diameter corresponding to k=0.6. From the data presented in Fig 2, this value is a good estimate for the 2017 data. But the 2016 values are lower probably more like 0.3. How much does that affect the D50 in your calculation and how much will that then change the NSMPS value.
- Other source for overcounting of a CCNC: sheath air filter lets particles pass. The artefact gets more severe with higher particle concentrations and cannot be corrected with a simple correction factor. Was the sheath air filter regularly checked? Or at least at the start and the end of the campaign?

Specific comments

1) Focusing on the data for 2017, where composition information is available, one can compare the k values from the CCNC ($k_{CCN}$) with the estimation from the composition data ($k_{chem}$). I used the simple ion pairing scheme (Gysel et al., 2007) and typical values for density and kappa (see Table 1) assuming that there are only the "usual" compounds (i.e., ammonium sulphate (AS), ammonium bisulphate (ABS), ammonium nitrate (AN), sulfuric acid (SA), organics (Org), and black carbon (BC)). To account for the range of $k_{org}$ I used 0.1 as a lower estimate and 0.2 as a reasonable upper value.
  Using the values I took "by eye" from Fig 2 in the manuscript for June 2017 (Case 1 in Table 2), I calculate a $k_{chem}$ of 0.367 ($k_{org}$=0.1) or 0.41 ($k_{org}$=0.2). $K_{chem}$ is of course only an estimation relying on several assumptions (e.g., the value for $k_{org}$ and density of organic, and that the ACSM composition is representative for the particles size range that activates). But the calculated $k_{chem}$ values are so much lower than the average $k_{CCN}$.

However, the monthly trends for 2017 are reflected in the ACSM composition. Changing only the contribution of SO4 and BC in the range that I estimated for the August 2017 case (BC doubles from 0.12 to 0.24 and SO4 decreases by the same amount), $k_{chem}$ decreases to 0.251 (Case 2). The amount of change (decrease by ~0.11) is similar to the decrease in $k_{CCN}$ from June to August 2017.

In Case 3 (Table 2), I created an example of how much the composition would have to change to get $k_{chem}$ in the range of 0.6. This is still assuming that there is no other compound in the particles. In this artificial case, the inorganic mass fraction is 0.75 and BC+organic are only 0.25. The particles are also shifted to the acidic regime (ABS and/or SA instead of AS) as ABS and SA have higher k values than AS. While this Case 3 is an arbitrary combination of numbers to reach k>0.6, it shows how much the SO4 to org+BC ratio would have to change to be in the right range for the $k_{CCN}$ values for August 2017. The Case 3 also doubles to illustrate how much the SO4 contribution would have to change to explain the difference between the 2016 values (avg $k_{CCN}$ 0.3 – 0.45) and the 2017 ones (avg $k_{CCN}$ 0.5 – 0.6). To repeat: to achieve a 0.2-0.3 increase in k, the SO4 contribution would have to change from 35% to 75%! I considered that an unlikely scenario for this location.

To summarise, I found it not possible to achieve the observed $k_{CCN}$ values of ~0.6 from the ACSM composition while using reasonable values for the parameters. I.e., the observed $k_{CCN}$ values cannot be explained by the species measured with ACSM (and BC). But adding a small fraction of NaCl (e.g., 20%) easily creates $k_{chem}$ values in the observed range.

Table 1: density and k values for the aerosol species/compounds

|     | density | kappa |
| --- | --- | --- |
| AS | 1.77 | 0.61 |
| ABS | 1.78 | 0.91 |
| AN | 1.72 | 0.67 |
| SA | 1.83 | 0.9 |
| Org | 1.3 | 0.1 or 0.2 |
| BC | 1.7 | 0 |

*Table 2: Mass fraction of ACSM species and volume fractions calculated using the ion pairing scheme and the values from Table 1.*

|     | Case 1 | Case 2 | Case 3 |
| --- | --- | --- | --- |
|     | mass fraction | mass fraction | mass fraction |
| SO4 | 0.35 | 0.23 | 0.625 |
| NO3 | 0.025 | 0.025 | 0.025 |
| NH4 | 0.1 | 0.1 | 0.1 |
| Org | 0.4 | 0.4 | 0.15 |
| BC | 0.12 | 0.24 | 0.1 |
|     | volume fraction | volume fraction | volume fraction |
| AS | 0.174 | 0.305 | 0 |
| ABS | 0.213 | 0 | 0.556 |
| AN | 0.029 | 0.028 | 0.031 |
| SA | 0 | 0 | 0.098 |
| Org | 0.475 | 0.459 | 0.193 |
| BC | 0.109 | 0.21 | 0.098 |
|     | $\kappa$chem | $\kappa$chem | $\kappa$chem |
| $\kappa_{org}$=0.1 | 0.367 | 0.251 | 0.66 |
| $\kappa_{org}$=0.2 | 0.414 | 0.300 | 0.675 |

The filter sample discussed later in the text prove that there are sea salt compounds (probably mostly NaCl) as well as K2SO4 and KNO3 (and potentially KCl) from BB. Potassium salts are also not quantifyable by ACSM. But due to the low number of available samples this data cannot prove or disprove their importance for the observed $k_{CCN}$. But the fact that there is a considerable amount of other inorganic species with very high hygroscopicity means that changes in their contribution need to be considered when interpreting the composition/hygroscopicity relationship.

2) Paragraph 44 – 55 and Line 362ff: "…changes in aerosol hygroscopicity become crucial in influencing cloud properties. ": hygroscopicity (described by k) is indeed very important for the CCN activity. But for the formation of cloud droplets, the particle size distribution also matters a lot. E.g. an aerosol population with a lower k can lead to more cloud droplets if the size of this population is shifted to have a larger number concentration at Dp>D50. How does this size effect compare to the k changes in the current data set? Or in other words, how much does the size distribution vary at this location? Any process that increases mass probably increases the average particle size and thus the ratio of particles with Dp>D50.
Dobracki et al. 2024 shows some average size spectra for clean and plume periods (Fig 11). Size distribution completely changes from being dominated by the smaller size mode at 30-40 nm to the second mode at 160 – 220nm. The effect of the size change on NCCN/CDNC will be much higher than the hygroscopicity change. Is this size change also linked to burning condition? How does the effect of the size change measure up vs the impact of a different k when estimating, e.g., Cloud droplet number concentration?

3) Methods section: More information about the used instrumentation is needed. The brand and models of the instruments should be named. This can go into a table in the supplement information.

4) Line 88: The SMPS size range in given in Zuidema et al. 2018 is 15-450nm. Which values are correct? This is relevant as later there is an extrapolation up to 1000nm

5) According to the report and some of the other publications there was also a UHSAS with a range of 50 – 1000nm. Why was that data not utilised?

6) Fig 2: The "error bars" in the box plot (10% and 90%) suggests that there were some results in 2017 with k almost =1. To my knowledge no atmospheric organic aerosol can reach such high values. Only inorganics like NaCl or H2SO4 have such high k. Looking at the corresponding size distributions in Dobracki 2024, this would mean almost all particles activated at SS=0.1% SS. I find that difficult to believe – could the authors please comment on this. Also combining this with the stronger "overcounting" of the CCNC in 2017 I get suspicious. The authors suggest that their correction method fixed the issue. But without knowing what was really causing the change from 2016 to 2017 I have some doubts about the robustness of the results.

7) Line 218ff: It is not clear which of the filters in (Dang et al., 2022) are utilised here. The description in Table 1 of Deng et al and in section 2.3 has some discrepancies. Deng et al lists 15 filters for 2017 not 17. I assume the 6 samples labelled "below cloud" are the MBL cases the authors used. There 4 were marine and 2 BB plumes – not evenly distributed. These two BB cases were the most aged emissions in the study (9 and 15 days from fire). All were collected between August 17th and September 7th. The ACSM data shows the lowest SO4 contribution for August 2017 and $k_{CCN}$ is lowest. I wonder how representative these measurements will be for the whole BB season. To reiterate: the interpretation is based on 2 filters collected on August 24th and Sept 2nd! From the observation that a significant portion of SO4 is linked to BB sources during those two days, the authors extrapolate that the observed monthly and yearly differences in $k_{CCN}$ are linked mainly to the changes in SO4 from BB sources. I find that very bold. Variations in sea salt contribution, changes in BC ratios, contribution of potassium salts are ignored in this interpretation.

8) Fig 2 and Fig 8 in (Dobracki et al., 2024) seem to not have data for the first week of September 2017. One of the two BB plume filters was collected on September 2nd. I do not know how far away the aircraft was from the station, i.e., how much later the airmass sampled by the aircraft would have arrived at Ascension Island. But this makes me wonder even more how the authors can base so much

of their interpretation on a filter sample that may not be very representative for the majority of their dataset.

9) Line 242: Does this really show that SO4 stems from BB? Or does it mean that SO4 is produced along the path of the BB particles before they get over the ocean? I am wondering if there are other not BB related SO4 sources on the African continent along the trajectories. I.e., for time periods with more BB influence, the air is coming from the region with these other sources and thus times with more BB contribution will also have more of these not directly BB related SO4 sources.

10) The assumption that the extrapolation of the $k_{CCN}$ vs OA mass fraction relationship yields $k_{org}$ is only strictly correct if no important aerosol constituent was missed in the calculation of OA mass fraction. As described above, it is highly likely that there is a considerable amount of inorganic compounds that ACSM&SP2 do not detect. Thus, the OA fraction is probably too high.

11) Paragraph 332- 340: Beside the previous point, this is another case where I find the authors quite bold in their interpretation of the data. The fast majority of data points has OA mass fractions of <0.6. This means a very "far" extrapolation towards OA mass fraction =1. Also the $R^2$ values are very low 0.21-0.24. How reliable is such a extrapolation in this case?

12) Fig 7b, Line 390 ff: Is the difference in wind significant? Is a 4% change enough to change the burning condition? How does this compare to the effect of the increase in easterly wind direction? I.e., less transport from BB source region to the measurement station.

Language & minor things

Overall the language and presentation is very good.

Line 22 "and thus a lower $\kappa$" verb is missing (reduced from the first part of the sentence does not work for the second part)

Line 44 the "where" in this sentence refers to SEA which is not the place where the aerosol is emitted.

Line 94 "which $\Delta$ denotes" "which" is the wrong word here

Line 98&138 both section are numbered as "2.2"

Eq 1: (Petters and Kreidenweis, 2007) clearly state that this specific equation is an approximation for k>0.2. The use of it is probably ok as most k values are indeed >0.2 in this study. But the authors should use wording that represents that this equation is an approximation.

Fig 1: What are the values on the x axis? NSMPS total or 31-1000nm?

Fig 2: why did the authors chose not to use the established AMS/ACSM colour scheme (with green for organic and dark blue for NO3)?

Fig 5 y axis label has $\Delta BC/\Delta CO$ but text uses $BC/\Delta CO$.

Fig 5 caption. "monthly percentages". That is not the best way to describe what this figure is showing.

Fig 5 b where do the total carbon values come from?

Fig 7a what is meant with mean wind/RH 2016? Mean over the observed months?

References

Dang, C., Segal-Rozenhaimer, M., Che, H., Zhang, L., Formenti, P., Taylor, J., Dobracki, A., Purdue, S., Wong, P.-S., Nenes, A., Iii, A. S., Coe, H., Redemann, J., Zuidema, P., Howell, S., and Haywood, J.: Biomass burning and marine aerosol processing over the southeast Atlantic Ocean: a TEM single-particle analysis, Atmos. Chem. Phys, 22, 9389–9412, https://doi.org/10.5194/acp-22-9389-2022, 2022.

Dobracki, A., Lewis, E., Sedlacek III, A., Tatro, T., Zawadowicz, M., and Zuidema, P.: Burning conditions and transportation pathways determine biomass-burning aerosol properties in the Ascension Island marine boundary layer, Atmos. Chem. Phys. Discuss., https://doi.org/10.5194/egusphere-2024-1347, 2024.

Gysel, M., Crosier, J., Topping, D. O., Whitehead, J. D., Bower, K. N., Cubison, M. J., Williams, P. I., Flynn, M. J., McFiggans, G. B., and Coe, H.: Closure study between chemical composition and hygroscopic growth of aerosol particles during TORCH2, Atmos. Chem. Phys., 7, 6131–6144, https://doi.org/10.5194/acp-7-6131-2007, 2007.

Petters, M. D. and Kreidenweis, S. M.: A single parameter representation of hygroscopic growth and cloud condensation nucleus activity, Atmos. Chem. Phys., 7, 1961–1971, https://doi.org/10.5194/acp-7-1961-2007, 2007.

Zuidema, P., Alvarado, M., Chiu, C., DeSzoeke, S., Fairall, C., Feingold, G., Freedman, A., Ghan, S., Haywood, J., Kollias, P., Lewis, E., McFarquhar, G., McComiskey, A., Mechem, D., Onasch, T., Redemann, J., Romps, D., and Turner, D.: Layered Atlantic Smoke Interactions with Clouds (LASIC) Field Campaign Report, 47, 2018.

---

## Author Comment (AC1)

**Item-by-Item Responses to Reviewers**

Haochi Che, Lu Zhang, Michal Segal-Rozenhaimer, Caroline Dang, Paquita Zuidema, Arthur J. Sedlacek III

We sincerely thank both reviewers for their time, effort, and insightful comments, which have greatly helped improve our manuscript. We have carefully considered all feedback and made the recommended revisions. In particular, we have significantly revised the manuscript to include detailed discussions on the role of sea salt, as suggested. Additionally, we included an acknowledgment in the manuscript to

10   recognize the reviewers' valuable contributions to improving the quality of this work.

Below, we provide detailed, itemized responses to each comment. In this response document, reviewer comments are highlighted in light **blue**, our responses are in **black**, and the corresponding revisions made to the manuscript are shown in **green**.

Given the significant role of sea salt identified in our analysis, we have revised the title of the paper to:

15   **Aerosol hygroscopicity over the South-East Atlantic Ocean during the biomass burning season: Part II – Influence of sea salt and burning conditions on CCN hygroscopicity**

The **Abstract** and **Conclusion & discussion** have also been revised to reflect the critical role of sea salt in shaping aerosol hygroscopicity. In the **Methods** section, a new subsection has been added to describe the approach used to estimate sea salt contributions. Two new subsections have been added to the

20   **Results** section: one focusing on the contribution of sea salt, and another distinguishing the respective roles of sea salt and biomass burning aerosols in determining overall aerosol hygroscopicity. Additionally, the subsection on organic aerosol hygroscopicity has been removed, as the measured organic mass fraction is biased due to the instrument's inability to detect sea salt. All revisions are clearly highlighted in the tracked version of the manuscript.
* * *
**Reviewer #1**

This study presented the long-term measurement results of particle number size distribution, CCN, chemical composition and hygroscopicity at an island observatory in Southeast Atlantic. It highlights how the biomass burning, sea salt aerosols influencing the particle hygroscopicity, as well the

30   relationship between meteorological parameters, providing valuable datasets and scientific insights in understanding the aerosol-cloud interaction in marine boundary layer. This paper is well prepared. However, more robust analysis and comprehensive data are needed to enhance the credibility of the conclusion. For example, as there are a lot assumptions made to derive κ values, why not use the chemical composition date to calculate a κ to constrain your method. More detailed comments are given

35   below:

Many thanks for your constructive feedback. We have carefully considered all of your suggestions and have addressed them in the revised manuscript.

Regarding the aerosol chemical composition-derived κ (κ$_{chem}$), the primary reason it is not directly comparable to the CCN-derived κ (κ$_{CCN}$) is that our aerosol chemical composition measurements, obtained using the ACSM, are limited to non-refractory bulk components. As a result, key refractory species such as sodium chloride (NaCl), which is highly hygroscopic, are not detected. As also noted by Reviewer #2, excluding NaCl from the κ$_{chem}$ calculation can lead to an underestimation relative to κ$_{CCN}$.

Nonetheless, we have used κ$_{chem}$ in our revised analysis to help estimate the sea salt (NaCl) mass fraction. A more detailed explanation of how κ$_{chem}$ was used in this context is provided in our response to Reviewer #2's first general comment.

1.  In section 2.1, please listed all the instrument model and manufacture in the experiment setup.

We have included this information in the method section. The updated content is provided below.

The CCN concentrations at fixed supersaturations were measured using a cloud condensation nuclei counter (CCNC, Droplet Measurement Technologies, model CCN-200) (Roberts and Nenes, 2005). Aerosol size distributions, ranging from diameters of 10 nm to 500 nm, were measured with a Scanning Mobility Particle Spectrometer (SMPS, TSI, model 3936). Concentrations of aerosol chemical components—including organics, sulfate, nitrate, ammonium, and chloride—were measured using a quadrupole aerosol chemical speciation monitor (Q-ACSM, Aerodyne Research). Refractive black carbon (BC) was measured using a Single Particle Soot Photometer (SP2, Droplet Measurement Technologies), and carbon monoxide (CO) concentrations were measured using the CO/N2O/H2O Analyzer (CO-ANALYZER, Los Gatos Research, model 098-0014). All instruments were calibrated and the data converted to standard temperature and pressure conditions.

2.  Line 100, the size range of SMPS is 10-500 nm, but you derive PNSD of 3-1000 nm by extrapolation, that will cause large bias especially on new particle formation cases. I think the comparison between 10-500 nm and 3-1000 nm is necessary. And with the large discrepancy caused by dp<10 nm and dp>500 nm, the reason should be analyzed and make sure if the extrapolation is reasonable. Otherwise, I recommend only 10-500 nm should be discussed.

We appreciate the reviewer for pointing this out. In fact, the particle number size distribution (PNSD) from 3 to 1000 nm was never used in our analysis. Instead, we used two PNSD ranges: 10–1000 nm and 31–1000 nm. The former was used initially to compare with CCN concentrations at 1% supersaturation in order to provide a general sense of how the two instruments compare. The latter was used in all subsequent analyses, including correcting for counting differences between the instruments and calculating κ.

While the SMPS is capable of measuring particles down to 10 nm, the PNSD in the 31–500 nm range is directly measured and not extrapolated. Only the size range from 500 to 1000 nm was extrapolated. This extrapolation was necessary for κ calculations, as we do not have size-resolved CCN measurements to directly determine the critical activation diameter. Instead, the activation diameter was inferred from the activation fraction and the available PNSD data.

We acknowledge that extrapolating beyond 500 nm introduces some uncertainty. However, we believe this uncertainty is minimal. As shown in Figure R1, which presents hourly averaged aerosol size

distributions measured by the Ultra-High Sensitivity Aerosol Spectrometer (UHSAS; Droplet
80    Measurement Technologies), there is no noticeable peak in aerosol concentration above the SMPS
upper detection limit (marked by the white dashed line at 500 nm). This indicates that there is no sudden
increase in aerosol concentrations beyond the SMPS range, and thus, the extrapolation is reasonable
and unlikely to significantly influence our conclusions.

[Figure]

85

[Figure]

Figure R1. Aerosol size distribution measured by the UHSAS during the fire seasons of 2016 and 2017.
The plot is based on hourly averaged data. The white dashed line at 500 nm marks the upper size limit
of the SMPS.

90    We have revised the manuscript accordingly, and the updated content is included below.

The aerosol number size distributions measured by SMPS expressed as dN/dlogD, were fitted with a
bimodal log-normal distribution. This fitted distribution was then extrapolated to obtain the integrated
aerosol number concentration ($N_{SMPS}$) from 10 to 1000 nm. Since the SMPS measurement range is
limited to 10–500 nm, the size distribution between 500 and 1000 nm was obtained through
95    extrapolation. To assess the validity of this extrapolation, we validated the results with measurements
from the Ultra-High Sensitivity Aerosol Spectrometer (UHSAS), which covers a broader size range

from 60 to 1000 nm. As shown in Figure S1 (Supplement), UHSAS data confirmed that aerosol concentrations above 500 nm were generally low and lacked distinct peaks, supporting the extrapolation.

100    3. In section 2.2, it shows the CCN at ss of 1% was 30% higher than SMPS, it the counting efficiency of CPC is 1?

The Model 3938 SMPS undergoes calibration before installation to characterize both the electrostatic classifier and the CPC. CPC calibration includes verifying the inlet flow rate and determining the size-dependent particle counting efficiency. The counting efficiency of the TSI 3772 CPC is a function of
105    particle diameter (and, to some extent, particle composition) and is determined following the calibration protocol described in Hermann et al. (2007).

During calibration, aerosol is generated using a tube furnace via the evaporation-condensation method, size-classified with a TSI Model 3080 Electrostatic Classifier and Model 3085 Nano DMA, and measured against a TSI Model 3068A Aerosol Electrometer. The CPC counting efficiency reaches
110    approximately 1 for particles larger than 15 nm.

   4. Figure 1. Would you please lower down the x- and y- axis to (0, 0), which helps to know if the blues dots at the bottom are zero. The zero data doesn't make sense.

Thank you for suggestion. We have excluded data points with very low aerosol concentrations (total
115    aerosol number concentration below 30 $cm^{-3}$) to ensure a more robust fitting. The updated figure is provided below.

[Figure]

Figure 1. Correlations between the CCN concentration measured at 1% supersaturation and the integrated aerosol concentration from the SMPS. Two distinct correlations were identified in the data
120    using the K-means clustering method, indicated by light blue and orange points. Type 1 (light blue) includes data from June, July, September, and October of 2016, while Type 2 (orange) includes data from August 2016 and June to October 2017. The blue and red lines represent linear regressions fitted to each correlation type, respectively.

5. As you mentioned you also have ACSM measurement, why don't you use chemical composition to estimate κ , or validate the method you modified?

This is primarily because the ACSM cannot detect NaCl, which makes a significant contribution to the overall κ. However, we included the $\kappa_{chem}$ calculation in our analysis to help estimate the potential contribution of NaCl. For further details, please refer to our response to Reviewer #2's first general comment.

In section 3.1, would you please give more detailed information about the mass fraction of sulfate corresponding to the extremely low and high κ , such as 0.3 and 0.78?

We binned the κ values from 0.25 to 0.75 using a bin width of 0.05 and calculated the mean and standard deviation of the sulfate mass fraction within each bin, as shown in Figure R2. The results show a clear positive correlation between sulfate mass fraction and κ. However, for κ values greater than 0.6, the sulfate mass fraction levels off, remaining within the range of 0.5 to 0.6. This suggests that sulfate alone cannot fully explain the highest κ values observed, indicating the likely contribution of other highly hygroscopic species, such as sea salt.

[Figure]

Figure R2. Observed sulfate mass fraction across different κ bins. Error bars represent the standard deviation within each bin.

We have included this figure in the Supplementary Information and revised the corresponding section in the main manuscript as shown below.

For the inorganic components, a plausible explanation for the observed differences in κ between 2016 and 2017 could be changes in sulfate mass fraction. As illustrated in Figure 2(b), sulfate aerosols constituted the primary inorganic component, accounting for approximately 39% of the total aerosol mass and around 80% of the inorganic aerosol mass on average during the BB season in 2017. Furthermore, as illustrated in Figure S10, the mean sulfate mass fraction increased with κ, ranging from

0.2 to 0.6, as $\kappa$ rose from 0.3 to 0.75. Given that sulfate aerosols have a relatively high $\kappa$ value of ~0.6 (Petters and Kreidenweis, 2007), changes in sulfate mass fraction might significantly affect the overall $\kappa$ observed on the island.

In addition to sulfate, sea salt (NaCl) is another key contributor that may help explain the high $\kappa$ values observed in 2017. Sea salt has a high hygroscopicity, with its $\kappa$ values reaching up to 1.5 (Zieger et al., 2017). Due to its refractory nature, NaCl cannot be directly measured by the Q-ACSM, making its contribution difficult to quantify. However, the observation of aerosol $\kappa$ values exceeding 0.6—the typical value associated with pure sulfate aerosols—throughout all months of the 2017 fire season provided strong indirect evidence for the presence of sea salt. Notably, the mean sulfate mass fraction under these high-$\kappa$ conditions remained between 0.5 and 0.6 (Figure S10), indicating that sulfate alone cannot account for the observed hygroscopicity. Given that NaCl is significantly more hygroscopic than sulfate, even modest increases in its mass fraction can lead to substantial changes in overall $\kappa$. Therefore, a slightly higher proportion of sea salt is a plausible explanation for the increased aerosol hygroscopicity observed in 2017.

6. It is recommend giving the time series or the means of CCN and PNSD, as well as BC, for the both campaign in 2016 and 2017, respectively, to make the readers easily understood the general aerosol background level. It could be provided in the supplementary materials.

Thank you for this valuable suggestion. We have plotted the time series of total particle concentration (CN) measured by the CPC ($\geq$10 nm), CCN number concentration at 0.1% supersaturation, BC mass concentration measured by the SP2, and the particle number size distribution (PNSD) measured by the SMPS, as shown in Figure S1 (in the Supplement). Additionally, we have updated our analysis using the latest version of the ACSM dataset and have removed BC data after September 1, 2017, due to inlet issues that affected the SP2 measurements.

These adjustments, along with the availability of BC data, are now clearly described in the revised manuscript.

Refractive black carbon (BC) was measured using a Single Particle Soot Photometer (SP2, Droplet Measurement Technologies), and carbon monoxide (CO) concentrations were measured using the CO/N2O/H2O Analyzer (CO-ANALYZER, Los Gatos Research, model 098-0014). All instruments were calibrated and the data converted to standard temperature and pressure conditions. Additionally, BC data after September 1, 2017, were omitted from the analysis due to an inlet issue that affected the accuracy of SP2 measurements.

Upon examining the time series plot, we identified a distinct pattern in the particle number size distribution (PNSD) that differs notably before and after September 1, 2016. Prior to this date, significantly higher concentrations of fine particles smaller than 20 nm were observed, as highlighted by the red frame in Figure S1(d). In addition, the SMPS detection limit appears to have changed after this point, with particles around 10 nm no longer being detected. This abrupt change likely reflects a shift in instrument performance rather than a real atmospheric phenomenon. Such an instrumental shift could also explain the two distinct correlation patterns observed between the CCN concentration at 1% supersaturation ($N_{CCN1\%}$) and the integrated aerosol number concentration from the SMPS ($N_{SMPS}$, 31–1000 nm).

[Figure]

Figure S1. Time series of (a) total particle concentration (CN) measured by the CPC, (b) CCN concentration at 0.1% supersaturation, (c) refractory black carbon (BC) mass concentration measured by the SP2, and (d) the particle number size distribution (PNSD) measured by the SMPS.

To investigate whether the two different correlation patterns (Type 1 and Type 2, as identified in Section 2.2 of the manuscript) observed between $N_{CCN1\%}$ and $N_{SMPS}$ were caused by changes in the SMPS, we analyzed their monthly correlations, as shown in Figure S2. A noticeable shift is evident during 2016: the correlation follows the Type 1 pattern from June to July, shifts toward Type 2 in August, and then reverts back toward Type 1 from September to October. The data in August appear relatively scattered, suggesting a transitional period with a mix of both correlation types. From November 2016 onward, the correlation remains consistently aligned with Type 2 through the end of the campaign.

This pattern suggests that the shift in the $N_{CCN1\%}$–$N_{SMPS}$ correlation cannot be solely attributed to changes in SMPS performance, which occurred in September 2016. Since the SMPS measurement differences were primarily confined to the smaller size range (10–20 nm), and $N_{SMPS}$ was integrated from 31 to 1000 nm, these differences should have a minimal impact on the integrated $N_{SMPS}$. Therefore, it is unlikely that the observed shifts in the $N_{CCN1\%}$–$N_{SMPS}$ correlation were driven entirely by SMPS-related changes. Instead, the results suggest that other instrumental factors—most likely associated with the CCN measurements—may have contributed to the emergence of the distinct correlation patterns.

[Figure]

Figure S2. Monthly correlations between the CCN concentration measured at 1% supersaturation ($N_{CCN1\%}$) and the integrated aerosol number concentration from SMPS ($N_{SMPS}$, integrated from 31 nm to 1000 nm) throughout the LASIC campaign. The blue and red lines indicate the two distinct correlation patterns (Type 1 and Type 2) identified and discussed in the main text.

To further investigate this issue and respond to Reviewer #2's question, we compared $N_{CCN1\%}$ with CN data measured by the CPC (lower cutoff diameter at 10 nm). The monthly correlations are presented in Figure S3.

[Figure]

Figure S3. Similar to Figure S2, but showing the correlations between $N_{CCN1\%}$ and CN concentration measured by the CPC (with a cutoff diameter of 10 nm).

The correlations between $N_{CCN1\%}$ and CN also shifted from Type 1 to Type 2 around November 2016, closely mirroring the pattern observed in the $N_{CCN1\%}$ - $N_{SMPS}$ relationship. This parallel behavior across both correlations suggests that the temporary shift in $N_{CCN1\%}$ - $N_{SMPS}$ observed in August 2016 was likely influenced by transient changes in the SMPS. However, the sustained shift beginning in November 2016 indicates a more permanent instrumental drift or alteration within the CCN counter, rather than an issue specific to the SMPS or CPC.

Overall, the analysis indicates that the sustained change in the correlation between $N_{CCN1\%}$ and $N_{SMPS}$, starting in November 2016, is primarily attributable to a drift or permanent change in the CCN counter. In contrast, the temporary deviation observed in August 2016 appears to be the result of short-term variability or changes in SMPS performance.

This discussion has been included in the Supplement, and the manuscript has been revised accordingly, as shown below:

Applying our modified approach, we observed two types of relationships between $N_{CCN1\%}$ and $N_{SMPS\_31}$ during the BB seasons in 2016 and 2017. We used K-means clustering to quantitatively categorize and separate the data, with the results shown in Figure 1. Type 1 includes observations from the 2016 BB season, excluding August, while Type 2 includes observations from the 2017 BB season and August 2016. Both types exhibit strong linear relationships with $R^2 \sim 0.98$. Although the underlying cause of these distinct relationships remains uncertain, the shift in correlation patterns may be related to instrumental changes in the CCN counter, as further discussed in Supplementary Note 2. After the two clusters were identified, we derived separate linear fits for each type and used the corresponding equations to scale the CCN data, thereby correcting for counting discrepancies between CCN and SMPS.

7.  As the observatory get more influence from sea salt aerosols in 2017, does the PNSD shifted to larger size as compared that of 2016?

Yes, this can be seen in Figure R1, which compares the PNSD between the two years with the UHSAS data. There was a clear increase in the large aerosols above 500 nm in 2017.

8.  In section 3.6, as the authors analyzed how the meteorological factors influencing the BB and sea salt transport, have you exclude the influence by precipitation?

Precipitation in the SEA region remained stable between 2016 and 2017, as investigated by Che et al. (2022). Figure 7 in their study shows that while precipitation varied slightly from month to month, there were no significant differences when comparing the same months across the two years. This indicates that the distinct $\kappa$ values observed between 2016 and 2017 were not influenced by changes in precipitation.

**Reviewer #2**

The authors present the monthly averages of the particle hygroscopicity from Cloud Condensation Nuclei activity measurements at Ascension Island Station in the Marine Boundary Layer of the Southeastern Atlantic for two consecutive Biomass Burning (BB) seasons (2016 & 2017). They explain the observed differences in the monthly mean values with changes in the aerosol particle composition,

namely the change in contribution of sulphate (SO4) species. They claim that these composition changes are mainly stemming from changes in the BB conditions on the African continent. The topic is suitable for this journal and of interest for the atmospheric science community. While I agree with the general direction of their conclusion (BB aerosol has a very important influence on the hygroscopicity in the SEA region), I am not convinced by their specific arguments and interpretation as I specify in my comments below. Some of the authors' claims appear quite bold to me and lack enough supporting evidence from the presented data. I recommend major revisions to present a more comprehensive interpretation of the data.

Thank you very much for your constructive feedback. We have carefully considered all of your suggestions, with particular focus on addressing your concerns regarding the contribution of sea salt to the variation in κ.

Major comments

1. Focussing on SO4 contribution to explain the trends – while ignoring NaCl from see salt for most of the discussion. This is my biggest concern about this manuscript. The authors use the change in SO4 contribution as the main argument to explain the observed trends within a year and between the two years. They do mention that NaCl from marine sources probably influences the hygroscopicity and even prove the presence of NaCl in Section 3.2. In the paragraph lines 299 – 304, they state that the main differences between flaming and smouldering burning conditions is indeed the amount of BB aerosol reaching the station. Thus, the burning condition determines the ratio between (more hygroscopic) marine and (very likely less hygroscopic) BB aerosol. For me, this is a key finding but the authors then completely omit the presence of NaCl from their arguments and continue to use the SO4 ratio as the main argument. NaCl will affect the observed hygroscopicity much more than SO4 related compounds as it has by far the largest k value of all included species and thus even a small amount will strongly change k. Since the ACSM cannot quantify NaCl, it is impossible to judge how much it is contributing at different times.

In short, I do not believe that the small change in SO4 contribution can explain the observed differences while ignoring the other constituents – especially NaCl which the ACSM cannot see. In Specific comment (1), I provide more details about my concerns using the approximation of k from the ACSM &SP2 composition to show my doubts about the SO4 fraction argument. The authors need to revise their chain of arguments and broaden their discussion to include the sea salt/NaCl impact and not just focus on SO4.

We sincerely appreciate the reviewer for highlighting this important aspect. We fully agree that sea salt plays a significant role in influencing the κ value. Since NaCl is a refractory species, it cannot be measured by ACSM, making it challenging to directly assess sea salt concentrations at the LASIC site. However, we acknowledge the reviewer's analysis that sulfate (SO$_4$) alone cannot fully explain the observed variations in κ.

Dedrick et al. (2022) derived sea salt aerosol properties by applying a Mie inversion method to scattering measurements from a nephelometer (NEPH) and aerosol size distributions from the UHSAS during the clean season of the LASIC campaign. Building on their results, we estimated sea salt mass concentrations by assuming a sea salt aerosol density of 2.16 g cm$^{-3}$ (corresponding to NaCl) and calculating aerosol volume using the mode diameter reported in their study. These results are shown in

305    Figure R3, alongside mass concentrations of other species measured by the ACSM and BC concentrations measured by the SP2.

[Figure]

Figure R3. Monthly aerosol mass concentrations: (a) measured by ACSM and SP2 for different species, 310    and (b) sea salt mass inferred from NEPH and UHSAS.

The figure shows that sea salt accounted for more than half of the total aerosol mass measured by the ACSM and SP2 during the clean season, confirming its substantial presence in the ambient aerosol. However, comparable data are not available for the fire season due to limitations in UHSAS 315    measurements under BB conditions. During the BB season, the high-power infrared laser in the UHSAS heats light-absorbing particles—such as brown carbon, tarballs, and black carbon—causing them to evaporate and shrink. This evaporation results in a significant underestimation of particle size, making the instrument less reliable in heavily polluted BB plumes (Howell et al., 2021).

Despite this limitation, it is reasonable to assume that sea salt concentrations did not decrease 320    significantly during the BB season. Therefore, the sea salt mass observed during the clean season can serve as a baseline for estimating its contribution under BB conditions. This supports the likelihood of a strong sea salt presence during the BB season and highlights its impact on aerosol hygroscopicity.

[Figure]

Figure R4. Monthly mean mass fraction of aerosol species during the fire seasons of 2016 and 2017. The blue frame highlights the mass fractions in 2016, while the orange frame highlights those in 2017. Note the mass of aerosols below 2.5 um ($PM_{2.5}$) was used.

To compare sea salt concentrations during the fire seasons of 2016 and 2017, we utilized MERRA-2 reanalysis data to calculate the monthly mean mass fractions of aerosol species in $PM_{2.5}$, as shown in Figure R3. The results indicate a clear increase in sea salt mass in 2017, with the most significant rise occurring in June and July. In contrast, the sea salt mass fraction in September remained approximately the same between the two years. This trend closely aligns with the κ values derived from CCN measurements (see Figure 2 in the manuscript). The largest differences in κ between the two years were observed in June and July, while the smallest change occurred in September. These findings suggest that variations in sea salt (NaCl) concentrations likely played a significant role in the observed interannual differences in κ.

According to the MERRA-2 reanalysis, sea salt mass accounts for more than half of the total aerosol mass ($PM_{2.5}$), highlighting its substantial presence in the atmosphere. Although the sea salt mass fraction is expected to be lower in our measured aerosols ($PM_1$), it is still anticipated to contribute significantly to CCN activity due to the high hygroscopicity of NaCl.

To estimate the contribution of sea salt (represented as NaCl) to the observed aerosol hygroscopicity, we performed a closure analysis by comparing the observed $κ_{CCN}$ with the theoretically calculated $κ_{chem}$. The calculation of $κ_{chem}$ followed the mixing rule described in Petters and Kreidenweis (2007), using the densities and κ values of individual aerosol species listed in Table R1.

| | $(NH_4)_2SO_4$ | $NH_4HSO_4$ | $NH_4NO_3$ | $NH_4Cl$ | OA | $H_2SO_4$ | NaCl | BC |
|---|---|---|---|---|---|---|---|---|
| Density ($g/cm^3$) | 1.77 (Haynes et al., 2016) | 1.78 (Haynes et al., 2016) | 1.72 | 1.53 | 1.4 | 1.83 Haynes et al., 2016) | 2.16 | 1.8 |

|  |  |  | (Haynes et al., 2016) | (Haynes et al., 2016) | (Alfarra et al., 2006) |  | (Haynes et al., 2016) | (Bond and Bergstrom, 2006) |
|---|---|---|---|---|---|---|---|---|
| κ | 0.61 (Petters and Kreidenweis, 2007) | 0.7* | 0.67 (Petters and Kreidenweis, 2007) | 1* | 0.1 (Zhang et al., 2024) | 0.9 (Petters and Kreidenweis, 2007) | 1.5* (Zieger et al., 2017) | 0 (Petters and Kreidenweis, 2007) |

Table R1. Density and κ value different aerosols species

*Note the κ values for $NH_4HSO_4$ and $NH_4Cl$ are not reported in Petters and Kreidenweis (2007). Therefore, we derived their theoretical values using the following equation (Schulze et al., 2020):

$$\kappa_i = \left(\frac{M_w}{\rho_w}\right)\left(\frac{\rho_i}{M_i}\right)v_i$$

where $M_w$ and $\rho_w$ are the molar mass and density of water, respectively, and $M_i$, $\rho_i$, and $v_i$ are the molar mass, density, and van't Hoff factor of the inorganic component. The van't Hoff factor is assumed to be 2.5 for $NH_4HSO_4$ and 2 for $NH_4Cl$. The κ of NaCl was too low in Petters and Kreidenweis (2007) and a revised value of 1.5 was used (Zieger et al., 2017).

The closure procedure involved the following steps:

1. We first derived the volume fractions of each species based on measured aerosol chemical composition, applying the simplified ion-pairing scheme proposed by Gysel et al. (2007).

2. The initial $\kappa_{chem}$ was calculated with volume-weighted contributions of all species and their corresponding $\kappa$ values. Since NaCl was not measured directly, its initial volume was set to zero.

3. We then computed the difference between the $\kappa_{CCN}$ and $\kappa_{chem}$, referred to as the $\kappa_{residual}$ ($\kappa_{residual} = \kappa_{CCN} - \kappa_{chem}$), to estimate how much sea salt was needed.

4. If $\kappa_{residual}$ exceeded 0.02, small increments of NaCl volume (1% of the total non-NaCl aerosol volume) were iteratively added. After each addition, $\kappa_{chem}$ was recalculated, and the $\kappa_{residual}$ was re-evaluated.

5. This iterative adjustment continued until the $\kappa_{residual}$ was reduced to 0.02 or less. The final amount of added NaCl was then considered the estimated sea salt contribution required to achieve closure with the observed $\kappa_{CCN}$.

[Figure]

Figure R5. Updated (a) aerosol mass concentration and (b) mass fraction during the fire season in 2017.

The time series of aerosol species, including the estimated NaCl contribution, is shown in Figure R5. A consistently high fraction of $H_2SO_4$ is observed throughout the fire season. This is due to the measured $NH_4^+$ concentration being insufficient to fully neutralize $Cl^-$, $NO^{3-}$, and $SO_4^{2-}$ ions. In the absence of measurements for other cations that could balance the excess sulfate, we assigned the unpaired $SO_4^{2-}$ to $H_2SO_4$. However, other forms of sulfate salts, such as $Na_2SO_4$, could also be present. To assess the potential impact of using sulfate salts instead of $H_2SO_4$ on the derived NaCl concentrations, we recalculated the aerosol composition by assigning all excess $SO_4^{2-}$ as $Na_2SO_4$. The results, also shown in Figure R6, indicate that the difference between the two assumptions is minimal. This is expected, given that the κ value of $H_2SO_4$ is 0.9, while that of $Na_2SO_4$ is 0.8. As a result, this choice has no significant effect on the derived NaCl concentrations from the κ closure analysis.

It is important to note that $H_2SO_4$ and $(NH_4)_2SO_4$ do not co-exist in the aerosol phase. Although Figure R4(b) may appear to show both species simultaneously due to dense data masking temporal separation. In the calculation, $H_2SO_4$ appears only when $NH_4^+$ is insufficient to fully neutralize $SO_4^{2-}$ in the form of $NH_4HSO_4$. Under these conditions, the excess $SO_4^{2-}$ is represented as $H_2SO_4$, and $(NH_4)_2SO_4$ is not present.

The results clearly demonstrate that sea salt is a substantial and persistent component of the aerosol population on Ascension Island (Figure R6). The consistently high fraction of NaCl confirms that the high $κ_{CCN}$ values observed during the fire season cannot be attributed to sulfate alone. Although the highest total aerosol concentrations were recorded in August, driven by BB influence—as indicated by increased fractions of BC and OA—sea salt remained a significant contributor, accounting for approximately 18% of the total aerosol mass during this peak BB period.

Compared with MERRA-2 reanalysis data (Figure R4), the NaCl mass fraction derived from the $κ_{CCN}$ closure is more than half lower. This difference is expected, as our retrieval is based on particles within

the PM$_1$ range, while the reanalysis data report sea salt concentrations for PM$_{2.5}$, in which the masss of larger sea salt particles is expected to be higher.

[Figure]

Figure R6. Same as Figure R5 but with excess sulfate represented as Na$_2$SO$_4$ instead of H$_2$SO$_4$.

[Figure]

Figure R7. Derived monthly mean (a) mass fractions of aerosol species and (b) percentage contribution of each aerosol species to the total $\kappa_{CCN}$ during the fire season in 2017.

[Figure]

Figure R8. Same as Figure R7, but with excess sulfate represented as $Na_2SO_4$ instead of $H_2SO_4$.

Sea salt and sulfate-bearing species (comprising $SO_4^{2-}$ containing particles, hereafter referred to as sulfate) are the two most important contributors to aerosol hygroscopicity observed on Ascension Island. Figure R7 (b) illustrates the monthly contributions of each species to $\kappa_{CCN}$. From June to August, NaCl contributed 43%, 42%, and 51% of the total $\kappa_{CCN}$, while sulfate contributed 46%, 47%, and 37%, respectively. This balance reflects combination of high intrinsic hygroscopicity of NaCl and the large mass fraction of sulfate, making both species equally important in determining overall aerosol hygroscopicity.

To test the sensitivity of this conclusion to assumptions about the chemical form of excess sulfate, we recalculated the species contributions assuming all excess sulfate is present as $Na_2SO_4$ instead of $H_2SO_4$. As shown in Figure R8, the resulting differences are minimal. Under this alternative assumption, NaCl contributions remain essentially unchanged from June to August, while sulfate contributions shift only slightly to 46%, 48%, and 38%, respectively.

Taken together, these findings confirm that both sea salt and sulfate are critical in shaping aerosol hygroscopicity during the fire season. Therefore, the observed differences in overall $\kappa_{CCN}$ between the two years are most likely driven by changes in the relative contributions of sea salt, sulfate, or both.

This raises a new question: Which species was the dominant factor driving the interannual difference in the observed $\kappa$? To address this, we estimated the potential sulfate mass fraction in 2016, assuming no change in sea salt levels. A strong linear correlation between BC/$\Delta$CO and sulfate mass fraction is shown in Figure 4 (b) of the manuscript. Here, the sulfate mass fraction was calculated as the ratio of observed sulfate to total measured aerosol mass (no NaCl), which minimizes the influence of sea salt and ensures that the sulfate fraction primarily reflects contributions from BB.

Using this correlation, we first estimated the sulfate mass fraction in sea-salt-free aerosols for 2016 based on observed BC/$\Delta$CO values. To extend this estimate to the total aerosol population (including NaCl), we assumed a constant sea salt mass fraction between 2016 and 2017 and applied the linear

relationship between sulfate mass fractions with and without NaCl (Figure R9). This approach allowed us to convert the estimated sulfate fractions from sea-salt-free aerosols into equivalent values for NaCl-containing aerosols, which could then be used to calculate aerosol hygroscopicity.

By comparing these estimated sulfate mass fractions between 2016 and 2017, we quantified the year-to-year differences. Multiplying the sulfate mass fraction differences by the hygroscopicity of sulfate ($\kappa = 0.6$) enabled us to estimate the potential contribution of sulfate variations to the observed differences in $\kappa$ between the two years.

[Figure]

Figure R9. Relationship between sulfate mass fraction in observed aerosols without NaCl (sea-salt-free) and estimated sulfate mass fraction in aerosols containing NaCl. Colors indicate data density, representing the number of data points within each grid cell (divided into 40×40 bins across the data range). Solid black lines show linear regression fits, with corresponding regression equations provided in the legend.

To establish plausible upper and lower bounds for the sulfate mass fraction estimates, we vertically shifted the original linear relationship between sulfate mass fraction and BC/$\Delta$CO (Figure 4b) to encompass the majority of data points (density > 200), as shown in Figure R10. The resulting sulfate mass fractions for NaCl-containing aerosols in 2016 are shown in Figure R11 (a). Averaged over June to August, the sulfate mass fractions ranged from approximately 0.10 at the lower bound, to 0.21 at the median estimate, and up to 0.33 at the upper bound. Although the estimates span a relatively wide range (~0.2) between the highest and lowest scenarios, all consistently exhibit a decreasing trend from June through August.

[Figure]

Figure R10. Relationship between BC/ΔCO and sulfate mass fraction. Colors represent data density, defined as the number of data points within each grid cell (50×50 bins across the data range). Solid black lines indicate linear regression fits, with corresponding equations provided in the legend. Dashed lines indicate lower and upper bounds of regression fits derived from data points exceeding a density threshold of 200.

[revised manuscript text omitted]

2.  Separating data from clean periods and BB plumes before averaging.

The authors present most of the data as monthly averages which are, of course, of general interest and should be reported to better understand the average hygroscopicity of this region. (Dobracki et al., 2024) provide an in-depth analysis of composition and optical properties of the 2017 part of the data set. In their Fig 1, they show that there are distinct plumes in the burning season of 2017. From the data in their Table 2, I calculated that the clean air periods account for 5% (September 2017) to 15% (July 2017) of the data. I assume the 2016 data will also show significant differences between the months and potentially also between the years.

This aspect (the frequency of BB plumes) is not covered in the discussion. Especially for the Section 3.3 and 3.4 it would be beneficial to separate the data from clean periods from the plume ones. In

630      Figs 4 and 5 the clean period data points are probably close to BC/∆CO=0 and are not distinguished from the smouldering data points with BC/∆CO<0.003. I would hazard a guess that a lot of the spread of the data points with BC/∆CO ~0 comes from having both smouldering BB plume data and clean air. Separating the clean and plume data may also improve the correlation (R^2^values are pretty low) and thus strengthen the conclusions drawn from these Figures.

635

Thank you for the suggestion. In Dobracki et al. (2025), clean conditions were defined as periods when the BC mass concentration was below 20 ng m⁻³. In our manuscript, however, BC/∆CO values were only calculated when BC concentrations exceeded 20 ng m⁻³, to minimize the influence of instrument noise under very clean atmospheric conditions. As a result, the 'clean' conditions defined by Dobracki

640 et al. (2025) were excluded from our analysis. This approach is consistent with our previous study on aerosol optical properties on Ascension Island, in which BC/∆CO was also used (Che et al., 2022). We have now included this clarification in the Methods section of the revised manuscript.

The ratio of black carbon to excess carbon monoxide (BC/∆CO) was calculated to assess BB combustion conditions, where ∆ represents the enhancement above background levels. Background CO

645 concentrations were determined monthly, defined as the 5th percentile of measured CO values for each month. The BC/∆CO ratio is unitless, as both BC and CO were converted to the same units of mass concentration. To minimize the influence of instrument noise under clean conditions, we only included data where BC mass concentrations were greater than 20 ng m-3 in the BC/∆CO calculations. As a result, clean atmospheric conditions were not considered in analyses involving BC/∆CO. A more

650 detailed discussion about BC/∆CO can be found in Che et al. (2022b).

However, there was an error in the calculation of BC/∆CO in our previous manuscript, where ∆BC— calculated in the same manner as ∆CO—was mistakenly used instead of the actual BC concentration. While this did not affect the overall conclusions, it slightly altered the magnitude of the BC/∆CO values and led to inconsistencies with those reported in Che et al. (2022b). In the current revision, we have

655 corrected this error and updated all relevant figures accordingly.

[Figure]

Figure 4. Relationships of BC/∆CO with (a) $\kappa$ calculated at 0.1% supersaturation and, (b) sulfate mass fraction. The black lines represent linear regressions, with the corresponding equations displayed in the

660    legend. The color scale indicates the data density, which is the count of the data in the gridded 50x50
       bins of the data range. Note that panel (a) includes data from both the 2016 and 2017 BB seasons,
       whereas panel (b) only includes data from the 2017 BB season. Data identified as clean conditions,
       defined by BC mass concentrations below 20 ng m$^{-3}$, have been excluded.

[Figure]

665

Figure 5. (a) Monthly distributions of BC/$\Delta$CO for the BB seasons of 2016 (blue) and 2017 (orange).
The boxes represent the 25th, 50th (median), and 75th percentiles, while the whiskers extend to the 10th
and 90th percentiles. (b) Relationship between BC/$\Delta$CO and the BC to TC (total carbon, the sum of OA
and BC) mass ratio during the BB season in 2017. The color scale in (b) indicates data density,
670    represented as the count of data points within gridded 50x50 bins of the data range. Only data with
concurrent CCN measurements are included to minimize sampling bias and ensure consistency across
the dataset.

       The clean versus polluted discussion, as suggested by the reviewer, primarily aims to distinguish the
675    contributions of sea salt and BB aerosols. In our study, we quantitatively estimated the contribution of
       sea salt during the BB season through a κ-closure analysis, as explained in our previous responses.

3.  I find the approach for the correction of the "counting error" problematic and need some more
    information and clarifications.
680    -    The authors only show data for the BB seasons (June – October in both years), but their
         description sounds like that there were continuous measurements for 18 months at the location.
         How does the rest of the data look like? Was there a continuous drift from Type 1 to Type 2?
         I.e., was the issue getting gradually worse with time or was there a sudden change? A gradual
         change would point towards drift in instruments and no connection to the aerosol characteristics.
685    A sudden change could point to some event affecting the instruments. If the difference seems
         chaotic (no clear trends) it is more likely that the discrepancy is linked to some aerosol property
         that changes over the months (e.g., special features of the size distribution, aerosol type).

Thank you to the reviewer for pointing this out. First, we acknowledge a typo in the manuscript: the
LASIC campaign lasted 17 months, not 18. We have corrected this in the revised manuscript.

The Layered Atlantic Smoke Interactions with Clouds (LASIC), a 17-month field observation campaign, was conducted from June 1, 2016, to October 31, 2017, on Ascension Island in the SEA to address these uncertainties.

Regarding the dataset, we confirm that continuous measurements were available from June 2016 to October 2017. A detailed discussion of the counting discrepancies between CCN and SMPS, as well as between CCN and CPC, is provided in our response to Comment 7 from Reviewer #1.

In summary, the analysis indicates that the persistent shift in the correlation between $N_{CCN1\%}$ and $N_{SMPS}$, beginning in November 2016, is most likely due to instrumental drift or a permanent change in the performance of the CCN counter. In the manuscript, we deliberately avoid referring to this as a "counting error," and instead use more neutral terms such as "counting discrepancy" or "difference."

- According to Zuidema et al. (2018), there were two more CPCs (cutoff at 10nm and 2.5nm) and a UHSAS at the station. How does the total number concentration from these instruments compare to SMPS and CCNC? Do they also become more different from 2016 to 2017?

We compared the total particle number concentration $N_{SMPS}$ from the SMPS with the total condensation nuclei (CN) concentration measured by the Condensation Particle Counter (CPC, cutoff at 10 nm), as well as the CCN concentration at 1% supersaturation $N_{CCN1\%}$ during the fire seasons of 2016 and 2017. A detailed analysis of these comparisons is provided in our response to Comment 7 from Reviewer #1.

In summary, two distinct types of correlations were observed between CCN and CPC, with a clear shift occurring in November 2016. This shift is consistent with the change observed in the correlation between $N_{CCN1\%}$ and $N_{SMPS}$, further suggesting a potential instrumental change or drift in the CCN counter during that time.

We did not compare UHSAS with CCN and SMPS for two reasons:

- The UHSAS has a lower detection limit of 50 nm, whereas the SMPS can measure particles as small as 10 nm. As shown in Figure S1, there is a local peak particle concentration around 30 nm. Excluding particles smaller than 50 nm would lead to an underestimation of the total aerosol concentration, introducing significant errors when comparing CCN concentrations at 1% supersaturation. As a result, it would not be possible to establish a reliable correction between CCN and intergated UHSAS.
- The data quality of UHSAS was flagged as "suspect" (see detail in : https://adc.arm.gov/discovery/#/results/instrument_class_code::uhsas/site_code::asi/start_date::2016-06-01/end_date::2017-10-31). This is likely related to known undersizing issues with the UHSAS, particularly during the BB season. The high-power infrared laser in the UHSAS can heat light-absorbing particles—such as brown carbon, tarballs, and black carbon—causing them to partially or fully evaporate, which leads to artificially small particle size readings (Howell et al., 2021). As a result, the UHSAS will have a biased aerosol size distribution (under-sized) when BB aerosols are present. Since κ calculations are highly sensitive to aerosol diameter, such undersizing can affect the accuracy of the derived κ values.

- How often were the instruments calibrated during the 18 months? (and which calibrations were performed?) - I assume there were regular SS calibrations with ammonium sulphate (AS) using the SMPS setup with the same CPC as during the measurements? Was the same concentration mismatch observed in these calibration measurements? Did the mismatch change between 2016

and 2017? Was there a size dependency? If the calibrations also showed the discrepancy, it is very likely that there is simply a discrepancy between the CPC and CCNC. TSI states that their CPC counting is +/- 20% directly after calibration. If the calibration data looked fine, there is potentially some other systematic reason for the "counting error" and that it is different in 2016 and 2017. If there was a size dependence in the AS data, there was probably a more complex issue than a simple counting error which probably cannot be fixed by a simple correction factor.

735

The CCN counter was calibrated prior to the start of the campaign and once again during the campaign, with both supersaturation and flow settings calibrated. The flow remained stable throughout the measurement period (see the response to the next question). To further evaluate the stability of the supersaturation, we examined the activated droplet size distributions, following the methodology of Moore and Nenes (2009) and Raatikainen et al. (2013). Specifically, we analyzed droplet size distribution samples measured at 0.1% supersaturation in June 2016 and June 2017—the same supersaturation used in the analysis presented in the manuscript. As shown in Figure 3, the droplet size distributions remained consistent over time, with a median diameter of approximately 1.2 µm, indicating stable supersaturation conditions. Therefore, supersaturation variability is unlikely to explain the observed shift in CCN counts.

740

745

[Figure]

Figure R12. Droplet size distribution at 0.1% supersaturation.

750

The instrument calibration was conducted by the Atmospheric Radiation Measurement (ARM) program of the U.S. Department of Energy; however, the raw calibration data are not available. As discussed previously, only the CCN concentrations exhibited a shift, while the CPC measurements remained

consistent with the SMPS data throughout the campaign. This consistency suggests that the observed shift in CCN concentrations was not due to the ±20% uncertainty associated with CPC measurements.

- The authors modify the approach by using an activation diameter corresponding to k=0.6. From the data presented in Fig 2, this value is a good estimate for the 2017 data. But the 2016 values are lower probably more like 0.3. How much does that affect the D50 in your calculation and how much will that then change the NSMPS value.

The D50 corresponding to $\kappa = 0.6$ is approximately 31 nm at 1% supersaturation, while for $\kappa = 0.3$ it is around 39 nm. This implies that the difference in $N_{SMPS}$ arises from the aerosol concentration integrated between 31 nm and 39 nm, which depends on the aerosol size distribution. Using the average size distribution observed during the 2016 fire season, we calculated the mean aerosol number concentration within this range (31–39 nm) to be approximately 17 particles cm$^{-3}$. If $\kappa = 0.3$ were used to define the activation cutoff, these 17 particles would be excluded from the $N_{SMPS}$ calculation. However, this exclusion has a negligible effect, as the total NSMPS (from 31 to 1000 nm) averages around 277 particles cm$^{-3}$.

To illustrate this, we used June 2016 as an example—selected for its abundant data and wide CCN variability—and compared the correlation between CCN concentrations and two different SMPS integrations: from 31 nm to 1000 nm (N31) and from 39 nm to 1000 nm (N39). As shown in Figure R13, both correlations are nearly identical, further confirming that the impact of this size range adjustment on $\kappa$ estimation is minimal.

[Figure]

Figure R13. Correlation between integrated SMPS and CCN at 1% supersaturation.

To further evaluate the impact of the cutoff diameter on the calculated $\kappa$ at 0.1% supersaturation, we computed the $\kappa$ values for June 2016 using two different methods. $\kappa_{31}$ represents the $\kappa$ value derived using N31, while $\kappa_{39}$ corresponds to the value derived from N39. As shown in Figure R14, there is a slight difference between $\kappa_{31}$ and $\kappa_{39}$, primarily in the higher $\kappa$ range, with $\kappa_{39}$ being slightly lower than

$\kappa_{31}$. Given that the $\kappa_{31}$ values in 2016 generally range from 0.2 to 0.4, this implies that using N39 would result in calculated $\kappa$ values between 0.19 and 0.32, with the difference typically being less than 0.08.

Considering this small difference and the need for consistency across the campaign for comparative purposes, we have chosen to continue using the N31 method.

[Figure]

785

Figure R14. Correlation between $\kappa_{31}$ and $\kappa_{39}$ in June 2016.

We therefore included the above discussion in the supplement, and revised the manuscript as below.

We then compared $N_{CCN1\%}$ with the integrated aerosol concentration $N_{SMPS\_31}$ from 31 to 1000 nm. This
790  approach reduces the likelihood of overestimating $\kappa$ by excluding nucleation-mode particles unlikely to activate, but it may still result in a slight overestimation, as particles smaller than 31 nm typically contain a higher organic fraction and may have $\kappa$ values lower than 0.6. However, sensitivity tests demonstrate that adjusting the assumed $\kappa$ value for calculating the initial diameter in the $N_{SMPS}$ integration has minimal impact on our results, as detailed in Supplementary Notes 1.

795

-   Other source for overcounting of a CCNC: sheath air filter lets particles pass. The artefact gets more severe with higher particle concentrations and cannot be corrected with a simple correction factor. Was the sheath air filter regularly checked? Or at least at the start and the end of the campaign?

800  The sheath flow filter was regularly checked throughout the campaign for water and dust. Any data affected by flow issues were flagged and removed. The CCN data used in this study have stable flow, as shown in Figure R15.

[Figure]

Figure R15. Sheath and sample flow of CCN counter.

805

Specific comments

1.  Focusing on the data for 2017, where composition information is available, one can compare the κ values from the CCNC ($\kappa_{CCN}$) with the estimation from the composition data ($\kappa_{chem}$). I used the simple ion pairing scheme (Gysel et al., 2007) and typical values for density and kappa (see Table 1) assuming that there are only the "usual" compounds (i.e., ammonium sulphate (AS), ammonium bisulphate (ABS), ammonium nitrate (AN), sulfuric acid (SA), organics (Org), and black carbon (BC)). To account for the range of $\kappa_{org}$ I used 0.1 as a lower estimate and 0.2 as a reasonable upper value.

810

Using the values I took "by eye" from Fig 2 in the manuscript for June 2017 (Case 1 in Table 2), I calculate a $\kappa_{chem}$ of 0.367 ($\kappa_{org}$ =0.1) or 0.41 ($\kappa_{org}$ =0.2). $\kappa_{chem}$ is of course only an estimation relying on several assumptions (e.g., the value for $\kappa_{org}$ and density of organic, and that the ACSM composition is representative for the particles size range that activates). But the calculated $\kappa_{chem}$ values are so much lower than the average $\kappa_{CCN}$. However, the monthly trends for 2017 are reflected in the ACSM composition. Changing only the contribution of SO4 and BC in the range that I estimated for the August 2017 case (BC doubles from 0.12 to 0.24 and SO4 decreases by the same amount), $\kappa_{chem}$ decreases to 0.251 (Case 2). The amount of change (decrease by ~0.11) is similar to the decrease in $\kappa_{CCN}$ from June to August 2017. In Case 3 (Table 2), I created an example of how much the composition would have to change to get $\kappa_{chem}$ in the range of 0.6. This is still assuming that there is no other compound in the particles. In this artificial case, the inorganic mass fraction is 0.75 and BC+organic are only 0.25. The particles are also shifted to the acidic regime (ABS and/or SA instead of AS) as ABS and SA have higher κ values than AS. While this Case 3 is an arbitrary combination of numbers to reach $\kappa_o$>0.6, it shows how much the SO4 to org+BC ratio would have to change to be in the right range for the $\kappa_{CCN}$ values for August 2017. The Case 3 also doubles to illustrate how much the SO4 contribution would have to change to explain the difference between the 2016 values (avg $\kappa_{CCN}$ 0.3 − 0.45) and the 2017 ones (avg $\kappa_{CCN}$ 0.5 − 0.6). To repeat: to achieve a 0.2-0.3 increase in κ, the SO4 contribution would have to change from 35% to 75%! I considered that an unlikely scenario for this location.

815

820

825

830

835 To summarise, I found it not possible to achieve the observed $\kappa_{CCN}$ values of ~0.6 from the ACSM composition while using reasonable values for the parameters. I.e., the observed $\kappa_{CCN}$ values cannot be explained by the species measured with ACSM (and BC). But adding a small fraction of NaCl (e.g., 20%) easily creates $\kappa_{chem}$ values in the observed range.

840 Table 1: density and k values for the aerosol species/compounds

|  | density | kappa |
|---|---|---|
| AS | 1.77 | 0.61 |
| ABS | 1.78 | 0.91 |
| AN | 1.72 | 0.97 |
| SA | 1.83 | 0.9 |
| Org | 1.3 | 0.1 or 0.2 |
| BC | 1.7 | 0 |

Table 2: Mass fraction of ACSM species and volume fractions calculated using the ion pairing scheme and the values from Table 1.

|  | Case 1 | Case 2 | Case 3 |
|---|---|---|---|
|  | mass fraction | mass fraction | mass fraction |
| SO4 | 0.35 | 0.23 | 0.625 |
| NO3 | 0.025 | 0.025 | 0.025 |
| NH4 | 0.1 | 0.1 | 0.1 |
| Org | 0.4 | 0.4 | 0.15 |
| BC | 0.12 | 0.24 | 0.1 |
|  | volume fraction | volume fraction | volume fraction |
| AS | 0.174 | 0.305 | 0 |
| ABS | 0.213 | 0 | 0.556 |
| AN | 0.029 | 0.028 | 0.031 |
| SA | 0 | 0 | 0.098 |
| Org | 0.475 | 0.459 | 0.193 |
| BC | 0.109 | 0.21 | 0.098 |
|  | $\kappa_{chem}$ | $\kappa_{chem}$ | $\kappa_{chem}$ |
| $\kappa_{org} = 0.1$ | 0.367 | 0.251 | 0.66 |
| $\kappa_{org} = 0.2$ | 0.414 | 0.3 | 0.675 |

   The filter sample discussed later in the text prove that there are sea salt compounds (probably mostly NaCl) as well as K2SO4 and KNO3 (and potentially KCl) from BB. Potassium salts are also not quantifyable by ACSM. But due to the low number of available samples this data cannot prove or disprove their importance for the observed $\kappa_{CCN}$. But the fact that there is a considerable amount of other inorganic species with very high hygroscopicity means that changes in their contribution need

   to be considered when interpreting the composition/hygroscopicity relationship.

Thank you very much for highlighting this important point. We acknowledge that NaCl played a major role in driving the observed differences in $\kappa$ between the two years. In response, we have significantly revised the manuscript and added two dedicated sections discussing the influence of sea salt. Please refer to our response to the first general comment for details.

2.  Paragraph 44 – 55 and Line 362ff: "…changes in aerosol hygroscopicity become crucial in influencing cloud properties. ": hygroscopicity (described by k) is indeed very important for the CCN activity. But for the formation of cloud droplets, the particle size distribution also matters a lot. E.g. an aerosol population with a lower k can lead to more cloud droplets if the size of this
   population is shifted to have a larger number concentration at Dp>D50. How does this size effect compare to the k changes in the current data set? Or in other words, how much does the size distribution vary at this location? Any process that increases mass probably increases the average particle size and thus the ratio of particles with Dp>D50.
    Dobracki et al. 2024 shows some average size spectra for clean and plume periods (Fig 11). Size
   distribution completely changes from being dominated by the smaller size mode at 30-40 nm to the second mode at 160 – 220nm. The effect of the size change on NCCN/CDNC will be much higher than the hygroscopicity change. Is this size change also linked to burning condition? How does the effect of the size change measure up vs the impact of a different k when estimating, e.g., Cloud droplet number concentration?

 Since the ACSM cannot measure sea salt—which plays a significant role in determining the overall aerosol hygroscopicity observed on Ascension Island—the correlation between organic aerosol (OA) mass fraction and measured $\kappa$ is biased, as the OA mass fraction excludes the contribution from sea salt. For this reason, we have removed the entire section titled *Changes in Organic Aerosol Hygroscopicity*, to which the reviewer's comment referred.

 We agree with the reviewer that aerosol size distribution is an important factor influencing CCN concentrations. However, the focus of this manuscript, as reflected in its title—*Aerosol Hygroscopicity over the South-East Atlantic Ocean during the Biomass Burning Season*—is on aerosol hygroscopicity rather than CCN number concentrations. While an increase in the number of large particles may enhance total CCN concentrations, it does not affect the calculated $\kappa$, which is primarily determined by aerosol
 chemical composition, as also noted by the reviewer. Therefore, a detailed discussion on the impact of size distribution on CCN is beyond the scope of this study.

3.  Methods section: More information about the used instrumentation is needed. The brand and models of the instruments should be named. This can go into a table in the supplement information.

885 Thank you for the reviewer's suggestion. We have updated the instrument model and manufacturer information in the Methods section accordingly. This point was also raised by Reviewer #1 in Comment 1, the details can be found in the response to that comment.

In addition, the models and manufacturers of the CPCs (not used in the main text) are the TSI Model 3772 (measuring particles from 10 nm to 3 μm) and the TSI Model 3776 (measuring particles from 3

890 nm to 3 μm). And the UHSAS used in this reply is from Droplet Measurement Technologies Inc.

4. Line 88: The SMPS size range in given in Zuidema et al. 2018 is 15-450nm. Which values are correct? This is relevant as later there is an extrapolation up to 1000nm.

The SMPS used in the LASIC campaign was capable of measuring aerosol diameters ranging from 10

895 to 500 nm, as detailed in the instrument description available at ARM website: https://www.arm.gov/capabilities/instruments/smps. This measurement range is also consistent with the description of other studies utilizing SMPS, such as Dobracki et al. (2024).

Thank you for pointing out this detail. The SMPS used during the LASIC campaign was capable of measuring aerosol diameters ranging from 10 to 500 nm, as specified in the instrument description

900 provided by the U.S. Department of Energy's Atmospheric Radiation Measurement (ARM) program: https://www.arm.gov/capabilities/instruments/smps. This size range is also consistent with the description used in recent studies utilizing the same dataset, such as Dobracki et al. (2024).

[Figure]

Figure R16: SMPS-measured aerosol size distribution during the fire seasons of 2016 and 2017. The

905 black line represents the median distribution, while the light blue shaded area indicates the 10th to 90th percentile range.

To verify this range in our dataset, we analyzed the SMPS particle size distribution during the 2016 and 2017 fire seasons. As shown in Figure R16, the available data spans from 11 nm to 461.4 nm, with

910 values outside this range recorded as NaN. However, upon further examination witht the partile size

distrition time series (see Figure R1), we observed that starting on September 1, 2016, particles below ~15 nm were no longer detected. This shift likely reflects a change in the SMPS detection settings or lower limit, which may explain why Zuidema et al. (2018) reported a measurement range of 15–450 nm.

915 Given this context, we have chosen to reference the broader size range (10–500 nm) consistent with the ARM instrument documentation and Dobracki et al. (2024).

5. According to the report and some of the other publications there was also a UHSAS with a range of 50 – 1000nm. Why was that data not utilised?

920 We did not use UHSAS for two reasons:

- The UHSAS has a lower detection limit of 50 nm, whereas the SMPS can measure particles as small as 10 nm. As shown in Figure S1, there is a local peak particle concentration around 30 nm. Excluding particles smaller than 50 nm would lead to an underestimation of the total aerosol concentration, introducing significant errors when comparing CCN concentrations at 1% 925 supersaturation. As a result, it would not be possible to establish a reliable correction between CCN and intergated UHSAS.
- The data quality of UHSAS was flagged as "suspect" (see detail in : https://adc.arm.gov/discovery/#/results/instrument_class_code::uhsas/site_code::asi/start_date::20 16-06-01/end_date::2017-10-31). This is likely related to known undersizing issues with the 930 UHSAS, particularly during the BB season. The high-power infrared laser in the UHSAS can heat light-absorbing particles—such as brown carbon, tarballs, and black carbon—causing them to partially or fully evaporate, which leads to artificially small particle size readings (Howell et al., 2021). As a result, the UHSAS will have a biased aerosol size distribution (under-sized) when BB aerosols are present. Since κ calculations are highly sensitive to aerosol diameter, such undersizing 935 can affect the accuracy of the derived κ values.

6. Fig 2: The "error bars" in the box plot (10% and 90%) suggests that there were some results in 2017 with k almost =1. To my knowledge no atmospheric organic aerosol can reach such high values. Only inorganics like NaCl or H2SO4 have such high k. Looking at the corresponding size 940 distributions in Dobracki 2024, this would mean almost all particles activated at SS=0.1% SS. I find that difficult to believe – could the authors please comment on this. Also combining this with the stronger "overcounting" of the CCNC in 2017 I get suspicious. The authors suggest that their correction method fixed the issue. But without knowing what was really causing the change from 2016 to 2017 I have some doubts about the robustness of the results.

945 Thank you for pointing this out. The whiskers in the original figure did not represent the 10th and 90th percentiles, and we have corrected the figure accordingly. In the updated version, the highest observed κ value is approximately 0.78, which strongly suggests a contribution from sea salt. Based on κ-closure analysis, we estimated the NaCl mass fraction in $PM_1$, with the median value around 15% and the 90th percentile reaching up to 35%. These results highlight the significant role of sea salt in contributing to 950 aerosol hygroscopicity and help explain the high κ values observed during the study period.

7. Line 218ff: It is not clear which of the filters in (Dang et al., 2022) are utilised here. The description in Table 1 of Deng et al and in section 2.3 has some discrepancies. Deng et al lists 15 filters for 2017 not 17.

955 A total of 14 samples were collected near Ascension Island during the CLARIFY campaign. The sample identifiers are: Gold1, Gold8, Gold9, Gold10, Gold11, Gold14, Gold15, Gold18, Gold19, Gold20, Gold21, Gold22, Gold23, and Gold24. Of these, six samples—Gold8, Gold9, Gold10, Gold14, Gold21, and Gold23—were selected for use in our analysis. This information has been included in the Supplement. The manuscript was also corrected.

960 There were 14 samples collected near the island, but only 6 taken inside the MBL were used for further analysis in this study. A detailed description of each sample is provided by Dang et al. (2022). A total of 231 particles were analyzed, and elemental mass fractions for individual particles were determined through EDX analysis. Based on back trajectory analyses, filters collected in the MBL were evenly mixed between BB and marine sources, ensuring a representative mix of aerosol sources in the study.
965 A list of detected elements, along with their mean mass fractions and standard deviations, is presented in Table S2 (Supplement).

8. I assume the 6 samples labelled "below cloud" are the MBL cases the authors used. Yes. There 4 were marine and 2 BB plumes – not evenly distributed. These two BB cases were the most aged
970 emissions in the study (9 and 15 days from fire). All were collected between August 17th and September 7th . The ACSM data shows the lowest SO4 contribution for August 2017 and kCCN is lowest. I wonder how representative these measurements will be for the whole BB season. To reiterate: the interpretation is based on 2 filters collected on August 24th and Sept 2nd! From the observation that a significant portion of SO4 is linked to BB sources during those two days, the
975 authors extrapolate that the observed monthly and yearly differences in kCCN are linked mainly to the changes in SO4 from BB sources. I find that very bold. Variations in sea salt contribution, changes in BC ratios, contribution of potassium salts are ignored in this interpretation.

The interpretation presented in our study is based on all six filter samples, not just two. Although four of the samples were labeled as marine in Dang et al. (2022), the measured BC concentrations during
980 the filter sampling period ranged from 0.1 to 1 $\mu g/m^3$. These concentrations are still higher than the clean condition threshold defined in Dobracki et al. (2025), suggesting that the aerosols sampled were not purely marine in origin. The classification in Dang et al. (2022) was based on seven-day back trajectories to distinguish between marine and BB air mass origins. However, this approach may not fully capture the mixing processes that occur when BB aerosols are entrained above the marine
985 boundary layer and transported downward through small scale motions. This interpretation is supported by Zuidema et al. (2018b), which showed that back trajectories arriving at Ascension Island in August 2016 primarily originated over the ocean, while the high BC concentrations observed during that month indicated the influence of BB aerosols despite the marine-dominated trajectories.

Regarding the primary source of aerosol hygroscopicity, we have revised the manuscript to reflect that
990 sea salt is now considered the dominant contributing factor.

995

9.  Fig 2 and Fig 8 in (Dobracki et al., 2024) seem to not have data for the first week of September 2017. One of the two BB plume filters was collected on September 2nd. I do not know how far away the aircraft was from the station, i.e., how much later the airmass sampled by the aircraft would have arrived at Ascension Island. But this makes me wonder even more how the authors can base so much of their interpretation on a filter sample that may not be very representative for the majority of their dataset.

Our analysis was not based on just two filter samples, as suggested by the reviewer—all six filter samples were used. We did not trace the specific air masses associated with each filter sample to determine whether they arrived at Ascension Island. Instead, we applied a statistical approach to analyze the aerosol particles collected across all six samples. During the sampling periods, BC concentrations ranged from 0.1 to 1 µg/m$^3$, indicating a mixture of both marine-dominated and biomass burning (BB)-influenced conditions.

1000

1005

1010

10. Line 242: Does this really show that SO4 stems from BB? Or does it mean that SO4 is produced along the path of the BB particles before they get over the ocean? I am wondering if there are other not BB related SO4 sources on the African continent along the trajectories. I.e., for time periods with more BB influence, the air is coming from the region with these other sources and thus times with more BB contribution will also have more of these not directly BB related SO4 sources.

We consider most of the BB-related sulfate to be secondary, formed through the atmospheric oxidation of SO$_2$, which is emitted during the combustion process and subsequently oxidized during transport. However, it is true that other anthropogenic sources across the African continent may also emit SO$_2$, which contribute to sulfate formation along similar transport pathways as BB aerosols. These sources may include industrial activity and fossil fuel combustion, and their contributions can coincide with BB emissions. Distinguishing between BB-related and non-BB anthropogenic sulfate is indeed challenging due to the lack of unique chemical tracers.

1015

Therefore, we have revised the manuscript to clarify the potential overlap of sources and our interpretation. The updated text now reads:

1020

Sulfate aerosols observed at Ascension Island likely originate from three different sources: transported BB emissions, other anthropogenic activities, and local marine sources (Hossain et al., 2024). The marine contribution to sulfate arises mainly from the oxidation of dimethyl sulfide (DMS) emitted from the ocean surface, while BB and anthropogenic sources contribute through the oxidation of sulfur dioxide released during combustion. Notably, much of the BB activity in Africa is itself anthropogenic, as evidenced by a pronounced Sunday minimum pattern—a clear signature of human influence tied to weekly activity cycles (Earl et al., 2015). During the fire season, BB emissions are expected to dominate over other anthropogenic sulfate sources. However, since both BB and non-BB anthropogenic sulfate are transported from continental regions and often arrive together, distinguishing their individual contributions remains challenging and requires additional research. For this study, we categorized all sulfate from BB and non-BB anthropogenic sources during the fire season as primarily of BB origin. This classification is supported by Figure 3 (a), which revealed a strong correlation between sulfate and potassium for BB-origin aerosols.

1025

1030

11. The assumption that the extrapolation of the kCCN vs OA mass fraction relationship yields korg is only strictly correct if no important aerosol constituent was missed in the calculation of OA mass fraction. As described above, it is highly likely that there is a considerable amount of inorganic compounds that ACSM&SP2 do not detect. Thus, the OA fraction is probably too high.

Yes, we agree with the reviewer. This entire section has been removed from the manuscript.

12. Paragraph 332- 340: Beside the previous point, this is another case where I find the authors quite bold in their interpretation of the data. The fast majority of data points has OA mass fractions of <0.6. This means a very "far" extrapolation towards OA mass fraction =1. Also the R^2^values are very low 0.21-0.24. How reliable is such a extrapolation in this case?

This entire section has now been removed from the manuscript, as we do not have NaCl measurement.

13. Fig 7b, Line 390 ff: Is the difference in wind significant? Is a 4% change enough to change the burning condition? How does this compare to the effect of the increase in easterly wind direction? I.e., less transport from BB source region to the measurement station.

We appreciate the reviewer's insightful comment. We agree that the ~4% change in mean wind speed observed over the BB source region between 2016 and 2017 is relatively small and not statistically significant. While we cannot definitively assess whether this slight reduction in wind speed was sufficient to alter combustion conditions, we note that the average wind speed over the land areas was already quite low (~1.96 m/s). In such low-wind environments, even small decreases could potentially influence the combustion regime, favoring more smoldering conditions over flaming, which could in turn affect aerosol composition and hygroscopicity.

We also acknowledge the reviewer's point regarding changes in wind direction. An increase in easterly winds could plausibly have reduced the efficiency of BB aerosol transport toward Ascension Island in 2017, leading to lower observed BB influence at the measurement site. We have added this interpretation to the revised manuscript to reflect both the potential influence of reduced wind speed on burning conditions and the impact of shifting wind direction on BB aerosol transport.

The mean wind field is shown in Figure 9(a). Easterly winds dominated over both the land and Ascension Island. During the 2016 BB season, the mean wind speed was approximately 7.6 m/s around Ascension Island and around 2 m/s over the BB source regions on land (excluding the ocean). In 2017, wind speed over the BB regions on land decreased slightly by about 0.08 m/s (Figure 9b), accompanied by an increase in westerly winds, which oppose the direction of plume transport. This shift in wind direction may have reduced the efficiency of BB aerosol transport toward Ascension Island, potentially decreasing the influence of BB aerosols at the site in 2017. In addition, wind speed can influence combustion efficiency: higher wind speeds promote flaming combustion by enhancing oxygen supply, although strong winds can also suppress flaming by increasing cooling (Santoso et al., 2019). Therefore, the slight (~ 4%) reduction in wind speeds during the 2017 BB season may have limited oxygen supply, promoting less flaming combustion. This interpretation is consistent with observed changes in BC/ΔCO. Together, the reduced wind speed and altered wind direction likely contributed to both modified

1075 combustion conditions and decreased BB aerosol transport, helping to explain the interannual differences in aerosol composition and hygroscopicity observed at Ascension Island.

In addition to changes in wind patterns over land, we also examined variations in wind speeds over the ocean near Ascension Island. Specifically, we analyzed area-averaged hourly wind speeds within a ±4° latitude/longitude region centered on Ascension Island, as defined in the main text. The statistical 1080 distribution of wind speeds during the BB seasons of 2016 and 2017 is presented in Figure 10 (a new figure added in the manscript).

[Figure]

Figure 10. Density distribution of area-weighted mean hourly wind speeds during the BB seasons of 1085 2016 and 2017, calculated for the region surrounding Ascension Island (blue box in Figure 9).

As shown in the figure, wind speeds in 2017 generally shifted toward higher values. Notably, the frequency of wind speeds exceeding 10 m/s more than doubled compared to 2016. Since sea salt aerosol production increases exponentially with wind speed, this enhancement in high-wind conditions likely 1090 contributed significantly to the elevated sea salt concentrations observed in 2017.

We have revised the manuscript accordingly as follows.

Sea-salt aerosol concentrations are strongly influenced by surface wind speed, following a power-law relationship (Ovadnevaite et al., 2012). In the region highlighted in blue in Figure 9(c), centered on Ascension Island and spanning ± 4 degrees in both latitude and longitude, mean surface wind speeds 1095 during the 2017 BB season increased by approximately 0.22 m/s, or about 3%, compared to 2016. Additionally, the surface wind direction became more southeasterly, indicating enhanced transport of marine air masses from the Southern Atlantic towards the island. These changes likely contributed to an increase in sea-salt aerosol mass in 2017. Although the overall increase in mean wind speed was

relatively small, the production of sea salt aerosols is highly sensitive to wind speed, particularly under higher wind regimes. Figure 10 shows that the frequency of wind speeds exceeding 10 m/s more than doubled in 2017, supporting a notable enhancement in sea salt emissions. Given the high hygroscopicity of sea salt, even modest increases in its mass fraction can significantly impact in overall aerosol hygroscopicity.

**Language & minor things**

Overall the language and presentation is very good.

Line 22 "and thus a lower $\kappa$" verb is missing (reduced from the first part of the sentence does not work for the second part)

The entire abstract is rewritten to reflect the role of the sea salt. Here is the new version of the abstract:

Biomass burning (BB) significantly influences cloud condensation nuclei (CCN) concentrations over the southeastern Atlantic; however, aerosol hygroscopicity ($\kappa$)—a key factor for CCN activation—remains poorly constrained during the BB season. This study investigates $\kappa$ variability using in situ measurements from Ascension Island during the 2016 and 2017 BB seasons. Results show substantial monthly variability, with $\kappa$ values lowest in August and increasing through October. On average, $\kappa$ was significantly higher in 2017 (~0.55) than in 2016 (~0.33), suggesting that the aerosols in 2017 were more hygroscopic and more easily activated as CCN. Sulfate and sea salt were the two dominant contributors to $\kappa$ and the primary drivers of its interannual variability. During the 2017 BB season, sulfate—the major inorganic component—accounted for ~34% of the submicron aerosol mass, while sea salt, estimated via $\kappa$-closure analysis, contributed ~17%. The higher $\kappa$ in 2017 was largely attributed to increased sea salt, likely driven by stronger marine winds. Approximately 67% of sulfate was linked to BB emissions. Variations in BB combustion efficiency, modulated by regional meteorology, influenced sulfate concentrations and consequently $\kappa$ values. Specifically, higher relative humidity and lower wind speeds over BB source regions in 2017 favored smoldering combustion, resulting in greater sulfate fraction. Overall, the observed interannual differences in aerosol hygroscopicity reflect the combined impacts of BB combustion characteristics and sea salt emissions, underscoring the critical roles of both BB and marine aerosol sources in regulating aerosol-cloud interactions over the southeastern Atlantic.

Line 44 the "where" in this sentence refers to SEA which is not the place where the aerosol is emitted.

Thanks for the reviewer, we have revised the manuscript accordingly. The updated sentence now reads:

BB is a major source of aerosol particles in the SEA, contributing large quantities of organic aerosol (OA) and black carbon (BC).

Line 94 "which Δ denotes" "which" is the wrong word here

Thank you for pointing out this, we have corrected the manuscript.

The ratio of black carbon to excess carbon monoxide (BC/ΔCO) was calculated to assess BB combustion conditions, where ΔCO represents the enhancement above background levels.

1140

Thank you for pointing this out. We have corrected the manuscript to ensure all subsections are properly numbered.

1145 Eq 1: (Petters and Kreidenweis, 2007) clearly state that this specific equation is an approximation for k>0.2. The use of it is probably ok as most k values are indeed >0.2 in this study. But the authors should use wording that represents that this equation is an approximation.

Yes, Eq. 1 is indeed only valid when κ is greater than 0.2. We have revised the manuscript to reflect this limitation. The updated sentence now reads:

1150 Once $D_c$ is identified, the hygroscopicity parameter κ can be calculated using the following equation (Eq. 1), which is an analytical approximation valid primarily for $κ > 0.2$ (Petters and Kreidenweis, 2007):

Fig 1: What are the values on the x axis? NSMPS total or 31-1000nm?

Thank you for pointing this out. The x-axis label should indeed be $N_{SMPS\_31}$, representing the number of
1155 aerosol particles integrated from 31 to 1000 nm. We have now corrected the x-axis label in the figure accordingly.

Fig 2: why did the authors chose not to use the established AMS/ACSM colour scheme (with green for organic and dark blue for NO3)?

1160 All figure related with aerosol chemical composition have been updated with the classic color scheme.

Fig 5 y axis label has ΔBC/ΔCO but text uses BC/ΔCO.

Thank you for the very careful and detailed review. The y axis should be BC/ΔCO, we have corrected the figure to ensure consistency with the manuscript.

1165

Fig 5 caption. "monthly percentages". That is not the best way to describe what this figure is showing.

We have revised the figure caption as:

Figure 5. (a) Monthly distributions of BC/ΔCO for the BB seasons of 2016 (blue) and 2017 (orange). The boxes represent the 25th, 50th (median), and 75th percentiles, while the whiskers extend to the 10th
1170 and 90th percentiles. (b) Relationship between BC/ΔCO and the BC to TC (total carbon, which is the sum of OA and BC) mass ratio during the BB season in 2017. The color scale in (b) indicates data density, represented as the count of data points within gridded 50x50 bins of the data range. Only data with concurrent CCN measurements are included to minimize sampling bias and ensure consistency across the dataset.

To maintain consistency, the captions for all figures containing similar box-and-whisker plots have been updated accordingly.

Fig 5 b where do the total carbon values come from?

Thank you for the comment. The definition of total carbon (TC) is provided in the main text, where we state: "*This conclusion is supported by Figure 5(b), which shows that the ratio of BC to total carbon (TC, sum of OA and BC) increases with BC/ΔCO.*" To avoid confusion, we have now added this definition to the figure caption as well. Please see the updated caption provided in our response to the previous comment.

Fig 7a what is meant with mean wind/RH 2016? Mean over the observed months?

Yes, the mean values shown in Figure 7(a) represent averages calculated over the BB season in 2016 (June to October), consistent with the time frame used in our data analysis. We have also updated the figure caption to clarify this.

Figure 9. Mean and difference in surface wind and relative humidity (RH) during the BB season (June–October). (a) Mean surface wind in 2016. (b, c) Difference in surface wind between 2017 and 2016 (2017 - 2016) over (b) the African continent and (c) the region surrounding Ascension Island. (d) Mean RH in 2016. (e) Difference in RH between 2017 and 2016 (2017 - 2016). The black boxes indicate the main BB source regions and the blue boxes represent the area centered on Ascension Island (±4° latitude and longitude). Box-averaged values are shown in the corresponding box colors. Note that in panels (d) and (e), the unit (%) refers to RH values, not relative percentage change.

---

## Author Response (AR2)

**Item-by-Item Responses to Reviewers**

Haochi Che, Lu Zhang, Michal Segal-Rozenhaimer, Caroline Dang, Paquita Zuidema, Arthur J. Sedlacek III

We sincerely thank both reviewers for their time, effort, and insightful comments, which have greatly helped improve our manuscript. We have carefully considered all feedback and made the recommended revisions.

Below, we provide detailed, itemized responses to each comment. In this response document, reviewer comments are highlighted in light **blue**, our responses are in **black**, and the corresponding revisions made to the manuscript are shown in **green**.

**Reviewer #1**

The paper has been improved significantly after addressing the reviewers' comments point to point. Regarding this, there are still few things I'm not sure I have understood well. Once these issues are concerned, this manuscript can be accepted for publication.

Many thanks for your feedback. We have carefully considered all of your suggestions and have addressed them in the revised manuscript.

1. Line 135-136, why the particle size of 20-35 nm is defined and addressed here? In my opinion, the aged biomass burning aerosols and sea salt aerosols should be larger.

In this section, we aim to estimate the critical particle diameter at which aerosols activate as CCN at 1% supersaturation. This critical diameter represents the smallest particle size expected to activate, allowing for a more accurate comparison between CCN and integrated aerosol concentrations measured by SMPS.

We determined this critical diameter using $\kappa$-Köhler theory. At 1% supersaturation, particles as small as 20 nm would require an unrealistically high $\kappa$ value (~2.24) to activate. Conversely, a critical diameter larger than 40 nm would suggest an too low $\kappa$ value (0.2). Such low $\kappa$ values do not align with the aerosol composition influenced by marine sources (e.g., DMS).

Therefore, we constrained the critical activation diameter within a realistic range of 20–40 nm. Within this range, we selected $\kappa = 0.6$ as representative of sulfate-dominated aerosols, leading to a calculated critical activation diameter of 31 nm. Sensitivity tests (detailed in Supplementary Notes 1) demonstrated that choosing a lower $\kappa$ value of 0.3—reflecting increased organic aerosol content in this size range—had negligible impact on our results.

In summary, we adopted $\kappa = 0.6$ as a representative aerosol hygroscopicity value, allowing us to define a justified critical activation diameter (31 nm) for integrating SMPS data and

comparing it with CCN measurements at 1% supersaturation. The manuscript has been revised accordingly to avoid confusion.

we assigned a representative $\kappa$ of 0.6 for particles in the lower part of the Aitken mode range (~ 20-40 nm)—similar to that of sulfate aerosols.

2. Line 175: This numbers of closure procedure steps should be revised to avoid confusion, as numbers 1–4 are already designated for primary headings.

Thanks for the comment. We've replaced the numeric labels for the closure-procedure steps with lettered labels (a–e) to avoid conflicting with the primary heading numbers.

3. Line 185, please clarify why the criterion of 0.02 is chosen?

We consider $\kappa_{CCN}$ and $\kappa_{chem}$ to be in agreement whenever $|\kappa_{CCN}-\kappa_{chem}|<0.002$, where 0.002 represents the uncertainty from calculations. Over the full campaign, $\kappa_{CCN}$ ranged from 0.2 to 0.8, so this threshold corresponds to roughly 1% of the lowest measured $\kappa$ value (0.2).

To clarify, we revise the text as the follow:

If $\kappa_{residual}$ exceeded 0.02 (equivalent to 1% of the lowest campaign-wide $\kappa_{CCN}$), small increments of NaCl volume (1% of the total non-NaCl aerosol volume) were iteratively added. After each addition, $\kappa_{chem}$ was recalculated, and the $\kappa_{residual}$ was re-evaluated.
* * *
**Reviewer #2**

They authors provide a thorough revision of their original manuscript. My two main points of criticism, the omission of the contribution of sea salt and the counting discrepancy between CCN counter and SMPS, have been addressed.

The counting discrepancy and its correction was another critical comment in my previous review. The clarification in the manuscript text and the additional information in the Supplement Notes provide the necessary background to understand how this instrument artefact is handled and what impacts can be expected on the reported kappa values. Further, the detailed monthly data, the comparison with other instruments measuring particle concentrations, and the analysis of the temporal evolution of the artefact show that while the instrument performance was suboptimal, the true differences between the seasons and the two years will still be visible above this bias. i.e., the authors showed that the observed differences cannot be explained by the instrument artefact.

The inclusion of the thorough analysis of the role of sea salt in addition to the biomass burning related effects has now shifted the conclusions of this study. While there is a lack of direct measurements due to the set of instruments, the authors go now to some length to estimate the sea salt concentration as well as one can. They also present ranges/uncertainties to show how their approximations/assumptions would affect the interpretation. This is highly commendable and leads to a much more robust manuscript where the interpretations and claims are now well

supported. The overall conclusion, that the CCN activity of aerosol particles in the SEA region are affected by the interplay of sea salt from the ocean and sulphate and organic aerosol mainly from biomass burning on the African continent are very useful for the atmospheric science community.

One remaining weak point lies with the unfortunate absence of composition measurements for 2016. Using the MERRA-2 reanalysis data can serve as a proxy to detect general differences. The findings highlight the need for direct NaCl/sea salt measurements even under BB dominate conditions in such marine environments.

In the new section 3.6, one new question arose for me:

The MERRA-2 data in Fig 7 shows that there are clear differences in sea salt fraction between 2016 and 2017. But in Section 3.6, the authors assume that the sea salt fraction is constant between the years. This seems to be a contradiction.

Once this very minor clarification has been provided, I recommend prompt publication.

We sincerely thank the reviewer for the thoughtful feedback and constructive comments, which have significantly improved this manuscript. We fully agree that, despite the dominance of biomass-burning aerosols, direct measurements of aerosol chemical composition—including sea salt—are essential in marine environments.

Regarding to Section 3.6: because no composition measurements are available for 2016, Section 3.6 relies on the 2017 regression between sulfate mass fractions with and without NaCl to infer the total (NaCl-containing) sulfate fraction for 2016. This implicitly treats the sea-salt fraction as unchanged from one year to the next. As the reviewer notes, the observations instead show higher sea-salt loadings in 2017. Extra sea salt in 2017 would dilute the sulfate fraction, so our method likely underestimates the 2016 sulfate fraction.

Most of the sea-salt, however, appears in the $PM_{2.5}$ size range, whereas our analysis is confined to $PM_1$ aerosol. The sea-salt change should therefore be less pronounced—and the associated bias smaller—in $PM_1$. Because the uncertainty bounds in Section 3.6 were derived from conservative upper and lower shifts of the 2017 sulfate–BC/$\Delta$CO line, we are confident that bias introduced by inter-annual sea-salt variability remains within the stated uncertainty range.

To clarify this, we have revise the text as below:

By comparing these estimated sulfate mass fractions between 2016 and 2017, we quantified the year-to-year differences. Multiplying these differences by the hygroscopicity of sulfate ($\kappa = 0.6$) enabled us to estimate the potential contribution of sulfate variations to the observed $\kappa$ differences between the two years. This procedure probably underestimates the 2016 sulfate fraction, because Figure 7 suggests sea-salt were higher in 2017, which would dilute sulfate in that year relative to 2016. Nevertheless, the sea-salt increase is less pronounced in $PM_1$ than in $PM_{2.5}$, so the bias in our $PM_1$-based estimate is likely within the uncertainty range evaluated below.